# Oxygen needs sulfur, sulfur needs oxygen: a relationship of interdependence

Hiroki Sekine [1✉], Takaaki Akaike [2] & Hozumi Motohashi [1,3✉]

## Abstract

**Oxygen and sulfur, both members of the chalcogen group (group 16 elements), play fundamental roles in life. Ancient organisms primarily utilized sulfur for energy metabolism, while the rise in atmospheric oxygen facilitated the evolution of aerobic organisms, enabling highly efficient energy production. Nevertheless, all modern organisms, both aerobes and anaerobes, must protect themselves from oxygen toxicity. Interestingly, aerobes still rely on sulfur for survival. This dependence has been illuminated by the recent discovery of supersulfides, a novel class of biomolecules, made possible through advancements in technology and analytical methods. These breakthroughs are reshaping our understanding of biological processes and emphasizing the intricate interplay between oxygen and sulfur in regulating essential redox reactions. This review summarizes the latest insights into the biological roles of sulfur and oxygen, their interdependence in key processes, and their contributions to adaptive responses to environmental stressors. By exploring these interactions, we aim to provide a comprehensive perspective on how these elements drive survival strategies across diverse life forms, highlighting their indispensable roles in both human health and the sustenance of life.**

**Keywords** NRF2; Pyridoxal 5'-phosphate; Vitamin B6; Supersulfides; Mitochondria
**Subject Category** Metabolism

## Introduction

In the primordial ocean, and especially within hydrogen sulfide ($H_2S$)- and hydrogen ($H_2$)-rich anoxic environments, life is once believed to have utilized an energy metabolism similar to those modern chemolithotrophic bacteria found in hydrothermal chimneys that use both molecules as electron donors (Martin et al, 2008). Primitive anoxygenic phototrophic bacteria are likely also to have used $H_2S$, which releases electrons more readily than water ($H_2O$), in a manner similar to that of extant purple sulfur bacteria

and green sulfur bacteria (Olson and Straub, 2016). Later in the history of life, cyanobacteria evolved using water as an electron source through an advanced photosynthetic system; this is thought to have led to the accumulation of molecular oxygen ($O_2$) in the atmosphere (Komiya et al, 2008). In turn, this accumulation enabled the evolution of aerobic organisms that use atmospheric $O_2$ for respiration as a terminal electron acceptor. This allows them to generate large amounts of energy due to highly positive reduction potential of $O_2$. Therefore, sulfur has played a central role in the energy metabolism of life that pre-dates the emergence of oxygen. However, despite having different geochemical histories, sulfur and oxygen have been found to be closely interconnected and mutually dependent in their biological functions (Fig. 1).

Recent research has revealed that sulfur-dependent energy metabolism remains robust in metazoans, such as humans and mice, which rely on oxygen respiration (Akaike et al, 2017; Alam et al, 2023). More specifically, sulfur metabolism is coupled with the mitochondrial electron transport chain, and is essential for oxygenic respiration. This evolutionary breakthrough was made possible by the emergence of supersulfides (details described later) and the identification of enzymes responsible for their de novo synthesis (Ida et al, 2014; Akaike et al, 2017). A new concept emerging from this discovery is that oxygen utilization is dependent on sulfur. The roles of supersulfides are now widely acknowledged, and their presence is providing answers to numerous long-standing and unresolved questions in biochemistry and biology.

$O_2$ is used in mitochondria for oxidative phosphorylation and in various metabolic reactions catalyzed by dioxygenases, monooxygenases, and oxidases. Beyond these enzymatic oxidation processes, non-enzymatic, unregulated oxidation (which is believed to contribute to organismal aging) also occurs in cells and tissues in our currently oxygen-rich environment (Cabiscol et al, 2014). Managing oxygen toxicity has been a critical challenge for aerobic organisms and has led to the evolution of antioxidant systems as defense mechanisms. Notably, sulfur plays a widespread and significant role in the protection against oxidative stress. In fact, some supersulfide species such as hydropersulfides and hydropolysulfides possess strong antioxidant properties, and serve as basal protection mechanisms against oxidative stress and electrophiles (Noguchi et al, 2023). In addition, the Kelch-like ECH-associated protein 1 (KEAP1)–NF-E2-related factor 2 (NRF2) pathway, an inducible response mechanism to oxidative stress and

[1]Department of Medical Biochemistry, Tohoku University Graduate School of Medicine, 2-1 Seiryo-machi, Aoba-ku, Sendai, Miyagi 980-8575, Japan. [2]Department of Redox Molecular Medicine, Tohoku University Graduate School of Medicine, 2-1 Seiryo-machi, Aoba-ku, Sendai, Miyagi 980-8575, Japan. [3]Department of Gene Expression Regulation, Institute of Development, Aging and Cancer, Tohoku University, 4-1 Seiryo-machi, Aoba-ku, Sendai, Miyagi 980-8575, Japan. ✉E-mail: hiroki.sekine.c2@tohoku.ac.jp; hozumi.motohashi.a7@tohoku.ac.jp

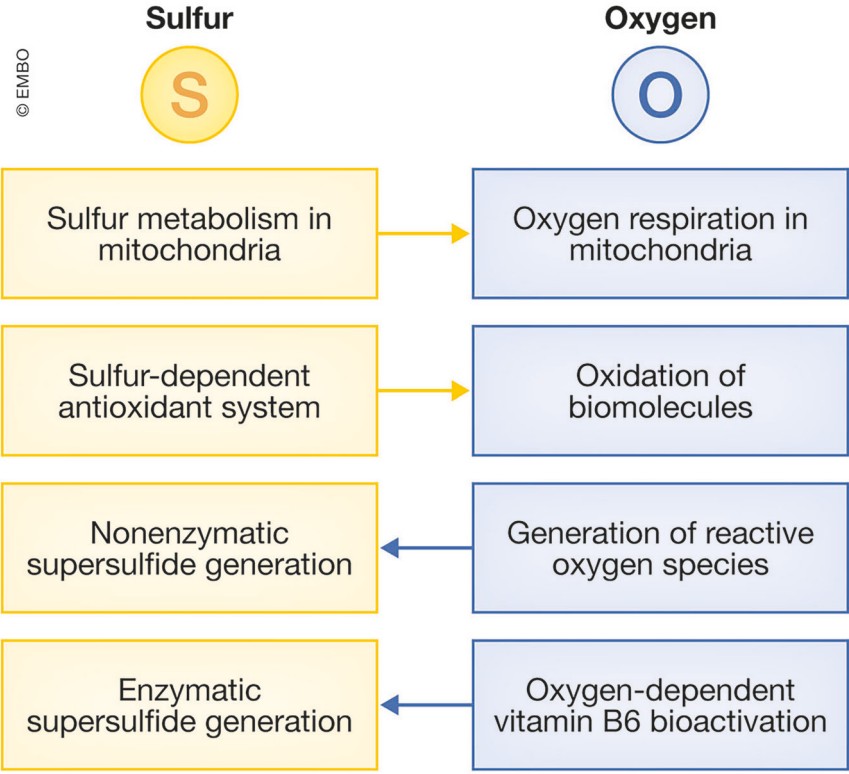

**Figure 1.** Overview of interdependence of sulfur and oxygen.

electrophiles, leverages sulfur reactivity in KEAP1-mediated electrophile sensing and NRF2-regulated enzymatic activities (Yamamoto et al, 2018).

Conversely, many pathological conditions are closely associated with hypoxic conditions where $O_2$-sensing and response mechanisms are active. The prolyl hydroxylase domain-containing protein (PHD)–HIF hypoxia-inducible factor (HIF) pathway is the most extensively studied response mechanism to hypoxia, with its dysfunction linked to various diseases (Schofield and Ratcliffe, 2004). Recently, a new $O_2$-sensing and response system, the pyridoxine 5'-phosphate oxidase (PNPO)–pyridoxal 5'-phosphate (PLP) pathway, has been identified, linking oxygen to sulfur metabolism (Sekine et al, 2024). All enzymes catalyzing the de novo synthesis of supersulfides require PLP as a cofactor, and PLP production is regulated by PNPO in response to $O_2$ levels.

This review will present the current understanding of the relationship between sulfur and oxygen across various biochemical, biological, and physiological processes. In addition, it will explore how two chalcogens, sulfur and oxygen, interactions affect cellular homeostasis, influence metabolic pathways, and contribute to the stress response, highlighting their significance to health and disease.

## Sulfur utilization by life

### Sulfur-utilizing microorganisms

Sulfur is an element that has driven the history of life on Earth since life first emerged in the primordial ocean (Mojzsis, 2007;

Kitadai et al, 2023). Alongside oxygen and selenium, sulfur belongs to Group 16 of the periodic table. Compared to oxygen, which many organisms use today, sulfur has a lower first ionization energy and higher electron affinity. This means that sulfur more readily releases and accepts electrons than oxygen does. In other words, the energy change associated with electron transfer is smaller with sulfur than with oxygen, making sulfur an element that organisms can readily use as a medium for redox reactions.

Deep-sea hydrothermal vents are regions of the seafloor where hot, anoxic, and chemically reducing water is released into the cold, and relatively oxygen-rich deep ocean. These environments can be categorized into four main habitats: hydrothermal chimneys, the subsurface surrounding vents, vent-associated fauna, and hydrothermal plumes (Dick, 2019). The microorganisms in each habitat exploit distinct metabolic profiles, depending on the redox conditions of their surroundings. For instance, in the aerobic subsurface that surrounds vents, bacteria utilize $H_2$ gas, elemental sulfur, or thiosulfate as electron donors, while oxygen serves as the electron acceptor. In contrast, bacteria inhabiting the anoxic, methanogenic hydrothermal chimneys rely on $H_2$ gas, methane, or other organic carbon compounds as electron donors, with elemental sulfur acting as the electron acceptor (Campbell et al, 2006; Dick, 2019). These hot, chemically reducing hydrothermal chimneys are thought to resemble the primordial conditions of early Earth, which were characterized by anaerobic, high-temperature environments rich in reduced chemicals such as sulfide, ferrous iron, $H_2$ gas, and methane (Dick, 2019). Modern microorganisms in hydrothermal chimneys, including thermophilic anaerobes that do not require $O_2$ as a terminal electron acceptor,

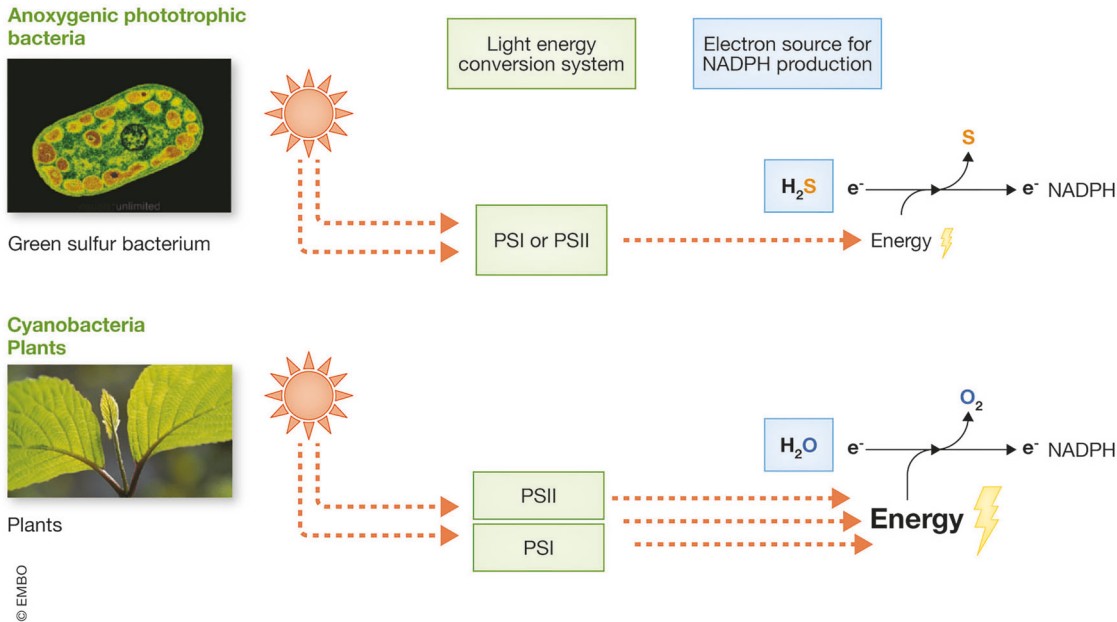

**Figure 2. Comparison of H₂S and water as electron sources for NADPH production in photosynthesis.**

Anoxygenic phototrophs have a single photosystem (PSI or PSII) and use H₂S to produce NADPH and elemental sulfur. Cyanobacteria evolved two PS systems (PSI and PSII) to use water to produce NADPH and oxygen, which was the basis for the evolution of aerobic life.

are believed to share key characteristics with some of Earth's earliest life forms (Martin et al, 2008; Wagner and Wiegel, 2008). While chemolithoautotrophic microorganisms utilize various electron donors and acceptors depending on the redox balance of their environment, sulfur plays an important role in their metabolism. Here, sulfur in the form of elemental sulfur, sulfide, and thiosulfate commonly serves as electron donors, with $O_2$ and nitrate functioning as electron acceptors. Conversely, sulfur in the form of elemental sulfur, sulfite, and sulfate acts as electron acceptors, utilizing $H_2$ gas, methane, and organic carbon compounds as electron donors (Inagaki et al, 2004; Nakagawa et al, 2005; Campbell et al, 2006; Jelen et al, 2018; Dick, 2019).

Phototrophic bacteria, which are classified into oxygenic cyanobacteria and anoxygenic phototrophic bacteria, offer a valuable example that illustrates the diversification of biochemical redox reactions. Oxygenic phototrophs, such as cyanobacteria and plants, possess two photosystems, PSI and PSII, whereas anoxygenic phototrophic bacteria have only a single photosystem, either PSI or PSII (Hohmann-Marriott and Blankenship, 2011). The two photosystems utilize different reaction centers for light-driven energy conversion: PSI donates electrons to iron–sulfur clusters, while PSII donates electrons to quinones. By linking these two photosystems, cyanobacteria have achieved a more efficient light-driven energy conversion that enables them to utilize water as an initial electron donor. This process, known as oxygenic photosynthesis, results in the production of $O_2$. In contrast, anoxygenic phototrophs, such as green sulfur bacteria and purple sulfur bacteria, lack the ability to extract electrons from water due to their single photosystem. Instead, they rely on alternative reductive compounds, such as $H_2S$, as electron donors (Frigaard and Dahl, 2009; Khasimov et al, 2021). Phylogenetic analyses strongly suggest that oxygenic photosynthesis evolved from anoxygenic

photosynthesis (Hanada, 2019). This implies that early phototrophic bacteria with a single photosystem may have more readily extracted electrons from $H_2S$ than from water. The transition to water oxidation is likely to have required the evolution of two interconnected photosystems, PSI and PSII (Fig. 2).

## Sulfur-sensing system

As discussed above, sulfur in the form of sulfide served as an electron donor during the early evolution of photosynthesis, and many existing photosynthetic bacteria still use sulfur compounds, such as $H_2S$, as a photosynthetic electron donor. SqrR, a sulfide-responsive transcriptional repressor, acts as a master regulator of sulfide-dependent gene expression in purple photosynthetic bacterium *Rhodobacter capsulatus* (Shimizu et al, 2017) (Fig. 3). Here, SqrR deficiency was found to upregulate the expression of almost half of the sulfide-responsive genes, including those encoding sulfide:quinone oxidoreductase, thiosulfate-sulfite oxidoreductases and thiosulfate sulfurtransferases, suggesting that SqrR plays a significant role in repressing a broad set of sulfur-responsive genes. SqrR belongs to the ArsR (arsenic repressor) superfamily, along with BigR (biofilm growth-associated repressor), which has been identified as another sulfur sensor (Fig. 3). In the pathogen *Acinetobacter baumannii*, BigR regulates a nonheme iron persulfide dioxygenase and membrane proteins involved in sulfur transport (YedE/YeeE) (Capdevila et al, 2021). Similarly, in the plant pathogen *Xylella fastidiosa*, BigR functions as a sulfur sensor that regulates the expression of gene products required for sulfur metabolism, including persulfide dioxygenase (Guimarães et al, 2011). While SqrR and BigR act as transcriptional repressors that lose their repressor activities in response to sulfide, FisR (Fis family transcriptional regulator) enhances transcriptional activation in response to sulfide, serving as another sulfur sensor in *Acinetobacter*

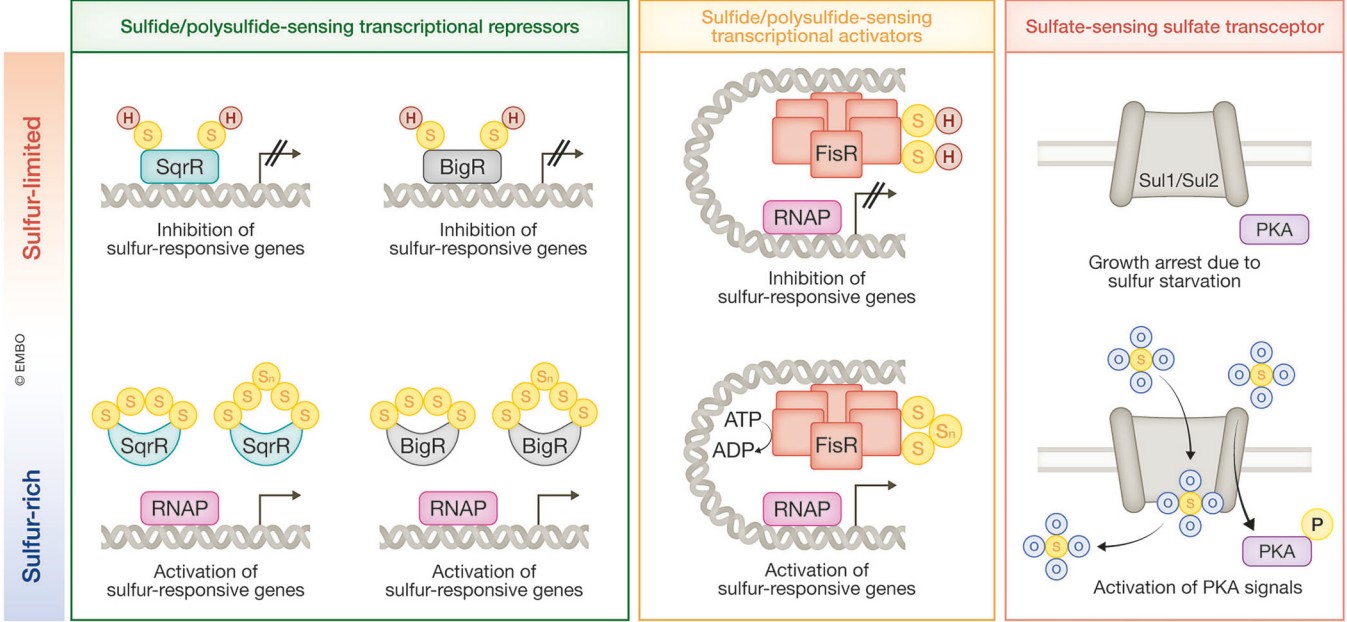

**Figure 3. Sulfur sensing in living organisms involves distinct mechanisms.**

Sulfide and polysulfides are detected by cysteine thiols within sensor proteins, while sulfate is recognized by sulfate transceptors, which serve as sulfate transporters for sulfate uptake and sulfate receptors for signal transduction. P phosphate residue, RNAP RNA polymerase.

*baumannii* (Walsh et al, 2020) (Fig. 3). These sulfur sensor proteins possess key cysteine residues that undergo formation of tetrasulfide or polysulfide bridges in sulfur-rich environments. These structural modifications induce conformational changes that inactivate the repressor functions of SqrR/BigR while enabling FisR to activate the transcription of sulfur-responsive genes by increasing its ATPase activity, activating RNA polymerization (Capdevila et al, 2021; Li et al, 2017; Shimizu and Masuda, 2020). In sulfur-rich conditions, persulfides are generated in cells, and their sulfur atoms are thought to be directly incorporated into the tetrasulfide/polysulfide bonds of SqrR/BigR and FisR. Thus, these proteins can be regarded as sulfur sensors (Capdevila et al, 2021; Li et al, 2017; Shimizu and Masuda, 2020).

Sulfur in the form of sulfate is an essential nutrient with significant regulatory effects on cellular metabolism and proliferation. In the yeast *Saccharomyces cerevisiae*, the sulfate transporters Sul1 and Sul2 have been reported to function as sulfate sensors (Kankipati et al, 2015) (Fig. 3). Sul1 and Sul2 also mediate the sulfate-induced activation of PKA (protein kinase A) signaling; this triggers the sulfate-induced exit from growth arrest after sulfur starvation. Intriguingly, transporter activities of Sul1 and Sul2 for sulfate uptake can be uncoupled from their sensor activities for downstream signaling. Although the precise molecular mechanisms through which Sul1 and Sul2 sense sulfate remain unknown, their ability to sense sulfate is likely to be dependent on their specific conformations (Kankipati et al, 2015).

## Enter supersulfides: a new paradigm for biomolecules

One unique feature of sulfur is its ability to undergo catenation: the formation of covalently-bonded long chains or rings. Sulfur is the only element that is thought to be capable of generating single-element-catenated, physiologically relevant metabolites. Whilst carbon is also able to form stable, long chains, sulfur chains are much less stable, perhaps leading to their relatively recent discovery of supersulfides as biomolecules. Recent advances in analytical chemistry have revealed that various molecular species with sulfur catenation are prevalent biomolecules across all organisms (Ida et al, 2014; Akaike et al, 2017). These molecules are now referred to as supersulfides, a category that includes persulfides and polysulfides (Fig. 4) (Akaike et al, 2024).

Unique chemical properties due to sulfur catenation distinguish supersulfides from other simple thiol-containing molecules. Perthiols (or hydropersulfides, RSSH) are more efficient nucleophiles than thiols (RSH) and are more readily deprotonated. For example, the pKa of glutathione persulfide (GSSH) is $5.45 \pm 0.03$, which is 3.49 units lower than that of glutathione (GSH) (8.94). This difference means that at pH 7.4, 2.8% of GSH exists in its deprotonated form, while 99% of GSSH is deprotonated (Benchoam et al, 2020). In addition, perthiolate anions (RSS$^-$) are much more efficient electron donors than thiolate anions (RS$^-$). Antioxidant-derived perthiyl radicals (RSS$\cdot$) are relatively inert radical species when compared to simple thiyl radicals (RS$\cdot$), making RSS less likely than RS to pose a threat of thiyl radical injury when generated within the cellular environment (Everett and Wardman, 1995).

Another important feature of supersulfides is that, when both RSSH and RSH are present, non-enzymatic transpersulfidation occurs, leading to a state of chemical equilibrium (Vasas et al, 2015; Fosnacht et al, 2024). Recognizing supersulfides as biomolecules requires understanding the equilibrium between metabolites in vivo, beyond enzyme-catalyzed biochemical reactions. Therefore, considering both enzymatic and non-enzymatic processes is crucial

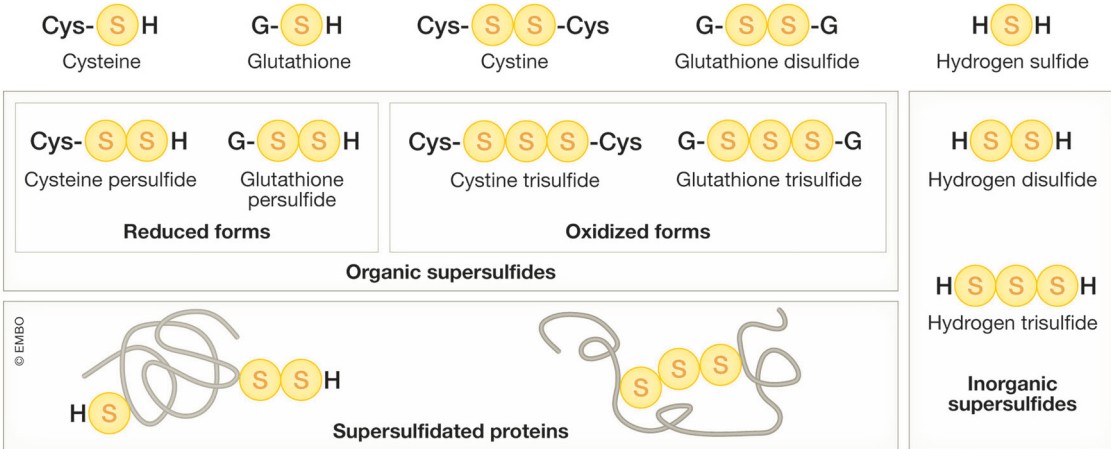

**Figure 4.** Various supersulfides in living organism, with corresponding thiol-containing metabolites shown at the top.

if we are to understand the diverse mechanisms by which supersulfides are generated. Cysteine persulfide (CysSSH) is enzymatically synthesized as a primary supersulfide from cystine by CBS (cystathionine-β-synthase) and CGL/CSE (cystathionine-γ-lyase) (Ida et al, 2014), and from cysteine by CARS (cysteine by cysteinyl tRNA synthetase) (Akaike et al, 2017) and MPST (3-mercaptopyruvate sulfurtransferase) in conjunction with CAT (cysteine aminotransferase) (Nagahara et al, 2018) (Fig. 5). Following this, non-enzymatic transpersulfidation from CysSSH is expected to produce a variety of supersulfides. In addition, the non-enzymatic formation of supersulfides from sulfide and thiols has been suggested to occur under oxidative stress (Paul and Snyder, 2015), also potentially generating diverse supersulfides. In addition to non-enzymatic transpersulfidation, MPST also catalyzes protein-protein transpersulfidation (Pedre et al, 2023). Of note, all enzymes involved in cysteine persulfide synthesis, except MPST, require PLP as a cofactor for their activities, suggesting a possibility of PLP-dependent nature of de novo supersulfide synthesis (see "PNPO-PLP system for connecting oxygen and sulfur").

Recent studies have shown that supersulfides are highly conserved across species, including both anaerobic and aerobic organisms, and play essential roles in fundamental biological processes such as energy metabolism, antioxidant activity, anti-inflammatory responses, and signal transduction (Akaike et al, 2024).

# Oxygen utilization by life

## Mitochondrial respiration

The emergence of oxygenic photosynthetic organisms, together with the major growth of continental crust and the consequent enhanced sedimentary deposition, are considered to have caused a large rise in atmospheric $O_2$ concentration (Komiya et al, 2008). Under this oxidative atmosphere, organisms developed the ability to utilize oxygen while simultaneously evolving defense mechanisms against its harmful effects. Higher eukaryotic aerobic organisms depend on oxygen for survival, yet oxygen poses significant risks to their existence

(Davies, 1995). This paradox arises from the unique electronic structure of $O_2$: each oxygen atom has one unpaired electron in its outer valence shell, and molecular oxygen contains two unpaired electrons, making atomic oxygen a free radical and molecular oxygen a bi-radical. While the concerted tetravalent reduction of $O_2$ by the mitochondrial electron transport chain (ETC) to produce water is relatively safe, univalent reduction generates reactive intermediates. In cellular environments, $O_2$ tends to undergo unscheduled univalent reduction, contributing to its potentially harmful effects.

Meanwhile, cells leverage reactive molecules derived from $O_2$ and oxidative reactions for physiological signal transduction (Finkel, 2011). Two primary sources of reactive oxygen species (ROS) are NADPH oxidases and the mitochondrial electron transport chain. NADPH oxidases generate the superoxide anion ($O_2 \cdot^-$) by transferring electrons from intracellular NADPH to extracellular $O_2$, using FAD and heme as electron carriers across the membrane (Sumimoto, 2008). In the mitochondrial ETC, electrons are transferred from NADH to $O_2$ within the mitochondrial matrix via complexes embedded in the inner mitochondrial membrane (Nolfi-Donegan et al, 2020). A four-electron reduction catalyzed by Complex IV of ETC converts $O_2$ into water. However, a small fraction of electrons interact non-enzymatically with $O_2$, resulting in the formation of $O_2 \cdot^-$ through one-electron reduction. Depending on the dose and cellular context, ROS can therefore act either as signaling molecules or as damaging agents.

Oxygen plays a fundamental role in the emergence of mitochondrial energy metabolism in aerobic organisms. It enables higher organisms efficiently to produce energy from organic substances. Mitochondria generate ATP, the primary fuel for cellular anabolic reactions, through oxygen-dependent respiration (Chandel, 2021). The oxygen-dependent respiration supports the efficient production of ATP by completely oxidizing nutrients such as sugars, lipids, and amino acids into carbon dioxide and water. The process occurs in two main stages: first, the generation of reducing equivalents (NADH and $FADH_2$), and second, the production of ATP by harnessing the free energy difference between the reducing equivalents and $O_2$.

In the ETC, NADH and $FADH_2$ act as typical electron donors, while $O_2$, with its highly positive reduction potential, serves as the

**Figure 5. Enzymes catalyzing the formation of cysteine persulfide from cysteine, a key process for the de novo synthesis of supersulfides.**

Enzyme names are highlighted in green. All enzymes, except MPST, are PLP-dependent.

terminal electron acceptor. The ETC operates based on the principle that each successive electron carrier has a higher standard reduction potential than its predecessor. Recent studies have revealed a more flexible and versatile flow of electrons within the ETC. For instance, under hypoxic conditions, fumarate can serve as the terminal electron acceptor, reducing ubiquinone to generate succinate (Spinelli et al, 2021). This process involves reverse electron flow from ubiquinone to succinate via complex II. Similarly, functional impairments in complex III or IV can alter the electron transport pathway, diverting electrons away at the level of ubiquinone.

Another example of this adaptability involves dihydroorotate dehydrogenase (DHODH), a rate-limiting enzyme in de novo pyrimidine synthesis. DHODH transfers electrons to the ETC via ubiquinone while converting dihydroorotate to orotate. Inhibition of DHODH promotes respirasome assembly by reducing metabolites downstream of pyrimidine nucleotides, such as CTP, which are required for phospholipid synthesis. This reduction leads to increased peroxisomal-derived ether phospholipid synthesis, enhancing respirasome assembly to support cell growth (Bennett et al, 2021). This illustrates the functional coupling of pyrimidine synthesis and ETC activity at ubiquinone.

Ubiquinone, therefore, serves as a critical hub in the ETC, integrating electrons from various metabolic pathways (Banerjee et al, 2022); pathways that include sulfide-quinone oxidoreductase (SQOR) (Jackson et al, 2012), ETF-quinone oxidoreductase (ETF-

QO) (Zhang et al, 2006), and glycerol 3-phosphate dehydrogenase 2 (GPD2) (Yeh et al, 2008), in addition to complex II (SDH: Succinate Dehydrogenase) and DHODH. The ETC can be viewed as an electron "trash bin," where electrons are efficiently funneled to $O_2$ (when it is available), making $O_2$ the ultimate "trash bin" for electrons (Fig. 6).

## Metabolism catalyzed by oxygenases and oxidases

Mitochondrial respiration is a primary driver of oxygen consumption, but various biochemical reactions also utilize $O_2$ as a substrate, catalyzed by oxygenases and oxidases (Malmström, 1982). Oxygenases are classified into two groups: monooxygenases and dioxygenases. Monooxygenases catalyze the incorporation of one oxygen atom into organic substrates, while dioxygenases insert both atoms of $O_2$. In both cases, oxygenases activate $O_2$ to facilitate the introduction of oxygen atoms into stable molecules. To achieve this, these enzymes require both an electron source and cofactors capable of single-electron chemistry. In contrast, oxidases transfer one or two electrons from a donor to $O_2$, resulting in the production of $O_2^{·-}$, hydrogen peroxide ($H_2O_2$) or water. In these reactions, $O_2$ acts as an electron acceptor.

From an evolutionary perspective, since its introduction into an anaerobic biosphere during the Earth's long history, oxygen has served as a major metabolic driving force, enabling the expansion of metabolic networks and the emergence of new metabolites and

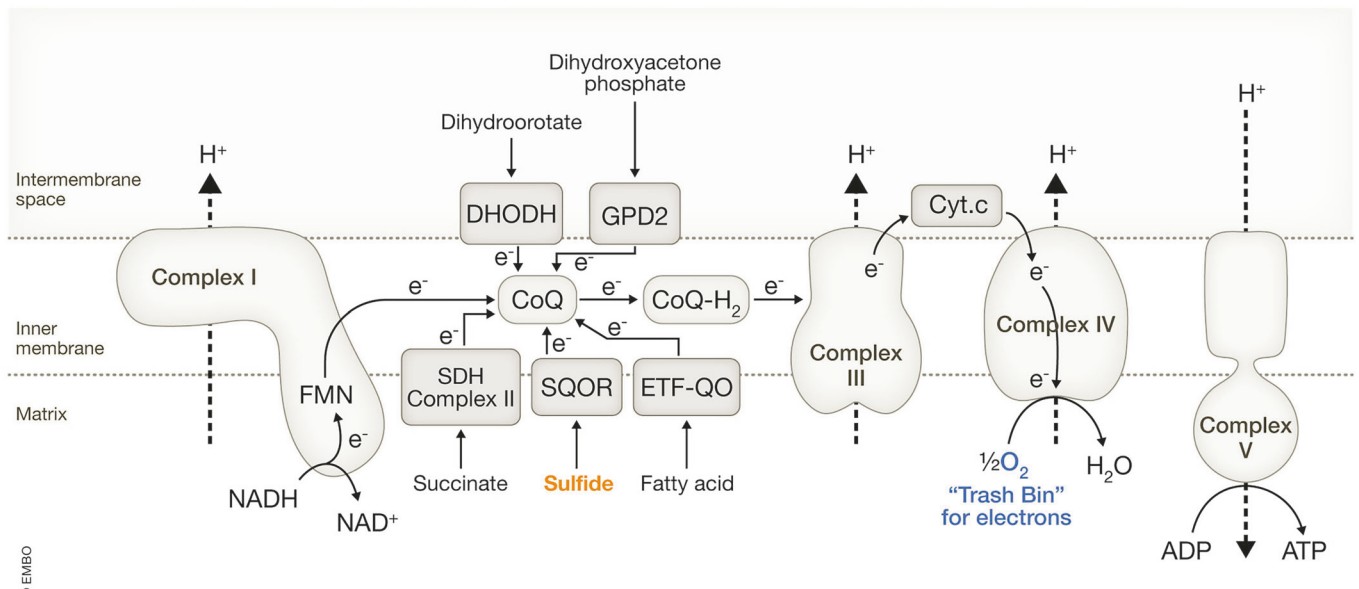

**Figure 6. O₂ as the ultimate "trash bin" for electrons.**

The mitochondrial electron transport chain (ETC) is depicted, consisting of Complex I (I), Complex III (III), and Complex IV (IV), along with ATP synthase (Complex V (V)). Various enzymes involved in oxidation reactions transfer electrons to the ETC via ubiquinone (CoQ), with O₂ serving as the terminal electron acceptor when available. FMN flavin mononucleotide, SDH succinate dehydrogenase, SQOR sulfide-quinone oxidoreductase, DHODH dihydroorotate dehydrogenase, ETF-QO ETF-quinone oxidoreductase, GPD2 glycerol 3-phosphate dehydrogenase 2.

enzymes. Enzymes involved in primary oxygen metabolism and ROS detoxification, such as catalase and superoxide dismutase, are present in both aerobes and anaerobes. However, enzymes like oxygenases and oxidases, that are key to the catabolism or anabolism of various substrates, including amino acids and xenobiotics, are found only in aerobes (Jabłońska and Tawfik, 2019). The availability of O₂ is likely to have driven an increase in metabolic network complexity, facilitated by secondary oxygen metabolism, larger than those found in anoxic networks (Raymond and Segrè, 2006). This complexity is mediated by the O₂-utilizing enzymes, oxygenases and oxidases.

The emergence of oxygenases and oxidases, which catalyze oxygen-dependent biochemical reactions, has enabled aerobic organisms to develop highly sophisticated metabolic processes essential for sustaining life and adapting to diverse environments while maintaining homeostasis. However, under hypoxic conditions, the activity of these enzymes is influenced by their affinity for O₂. Enzymes with higher $K_m$ values are more susceptible to inhibition in hypoxic environments, whereas those with lower $K_m$ values remain relatively unaffected by reduced O₂ availability.

## Oxygen-sensing system

The $K_m$ values for O₂ among enzymes that require O₂ for their activity vary significantly (Lee et al, 2023). Enzymes with higher $K_m$ values are more sensitive to depletions in O₂ availability and often function as O₂ sensors, most notably oxygenases. A well-known hypoxia response mechanism, the PHD-HIF pathway, capitalizes on the high $K_m$ values of PHD family proteins, which are 2-oxoglutarate (2-OG)-dependent dioxygenases (Jaakkola et al, 2001; Hirsilä et al, 2003). HIF, composed of α and β subunits, serves

as a master transcription factor for hypoxic responses. It regulates metabolic adaptation by promoting glycolysis, enhancing vascular formation, inducing erythropoiesis, and supporting cell survival through anti-apoptotic functions. Under normoxic conditions, PHD1, PHD2, and PHD3 hydroxylate proline residues on HIFα subunits (HIF1α and HIF2α), targeting them for proteasomal degradation. The high $K_m$ values of PHD proteins for O₂ allow their activity to be easily inhibited under hypoxic conditions, leading to stabilization of HIFα subunits and induction of the hypoxic response. Another example of 2-OG-dependent dioxygenases with high $K_m$ values for O₂ is histone demethylases, which are key epigenetic regulators. For instance, KDM6A (Lysine Demethylase 6 A), an H3K27 demethylase, has high $K_m$ values for O₂, enabling O₂ levels to regulate chromatin states and, consequently, control cell differentiation and fate (Chakraborty et al, 2019). Similarly, members of the KDM4 (Lysine Demethylase 4) family, such as KDM4A and KDM4E, which act as H3K9 demethylases, exhibit high $K_m$ values comparable to KDM6A, making them sensitive to O₂ availability (Cascella and Mirica, 2012; Hancock et al, 2017).

Some 2-OG-dependent dioxygenases exhibit O₂ sensitivity despite having lower $K_m$ values for O₂ compared to typical O₂ sensors. For example, KDM5A (Lysine Demethylase 5 A), an H3K4 demethylase, is inhibited under hypoxic conditions (Batie et al, 2019), and TET (Ten-Eleven Translocation) proteins, which mediate DNA demethylation, are also inactivated in hypoxia (Thienpont et al, 2016; Nishikawa et al, 2021). However, the $K_m$ values of KDM5A and TET proteins (TET1 and TET2) for O₂, 90 μM (Chakraborty et al, 2019) and 30 μM (Laukka et al, 2016) respectively, are significantly lower than those of PHD proteins (230–250 μM), KDM6A (200 μM), and KDM4 proteins

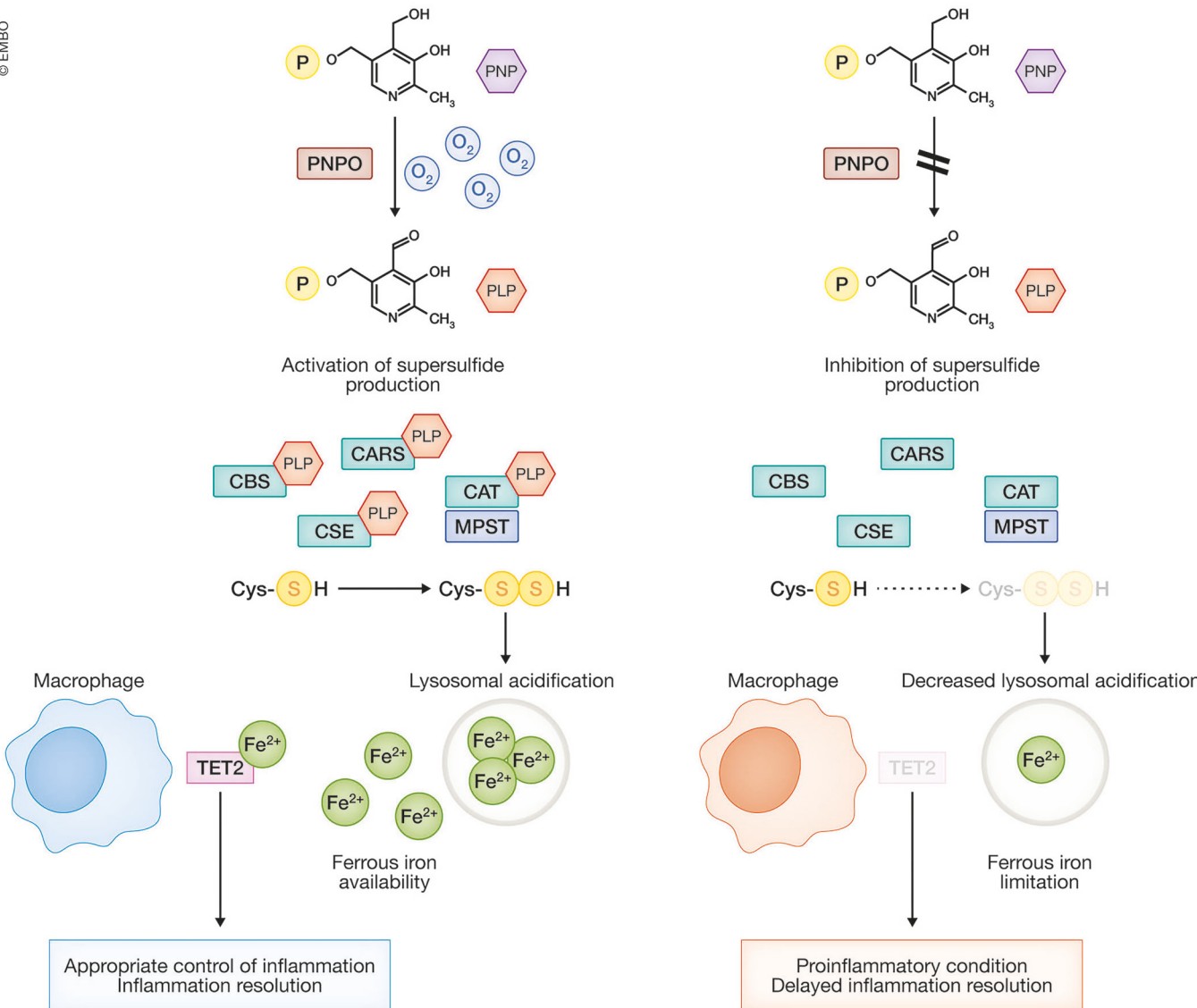

**Figure 7. The PNPO–PLP pathway promotes supersulfide production.**

PNPO acts as an $O_2$ sensor by ceasing its enzymatic activity under hypoxic conditions. Prolonged hypoxia leads to a gradual decrease in PLP, inhibiting PLP-dependent enzymes. Cysteine persulfide-synthesizing enzymes are particularly sensitive to reduced PLP levels, resulting in decreased supersulfides and impaired lysosomal acidification. Since lysosomal acidification is essential for ferrous iron availability, iron-dependent processes are affected under PLP-limited conditions. In macrophages, iron limitation depletes TET2 protein, which is critical for resolving inflammation, thereby promoting a proinflammatory phenotype under prolonged hypoxia. P phosphate residue.

(170–200 µM) (Hirsilä et al, 2003; Chakraborty et al, 2019; Cascella and Mirica, 2012; Hancock et al, 2017). In addition to direct regulation by $O_2$, our recent discovery of the PNPO–PLP pathway as a novel $O_2$-sensing system provides an alternative interpretation of how these dioxygenases respond to hypoxia (Sekine et al, 2024).

PNPO converts pyridoxine 5'-phosphate (PNP) and pyridox-amine 5'-phosphate (PMP) to PLP, the active form of vitamin B6, using $O_2$ as an electron acceptor to generate $H_2O_2$. PNPO activity is suppressed under hypoxic conditions, leading to a gradual decrease in cellular PLP concentration and an inhibition of PLP-dependent enzymatic reactions (Sekine et al, 2024). Compared with the rapid response of the PHD-HIF pathway, where HIF proteins are

promptly stabilized under hypoxia and initiate transcription, the PNPO–PLP pathway responds only to chronic hypoxia. Under short-term hypoxia, PLP levels remain unchanged despite PNPO inhibition, likely because PLP has a relatively long half-life.

We found that, among various PLP-dependent enzymes, supersulfide-synthesizing enzymes are particularly sensitive to PLP depletion under chronic hypoxia, resulting in a subsequent limitation in the cellular availability of supersulfides (Sekine et al, 2024). Under prolonged hypoxia, the decrease in supersulfides due to PLP insufficiency inhibits lysosomal acidification: a process that is essential for maintaining intracellular ferrous iron levels (Weber et al, 2020). Prolonged hypoxia therefore results in a decline in ferrous iron (Fig. 7).

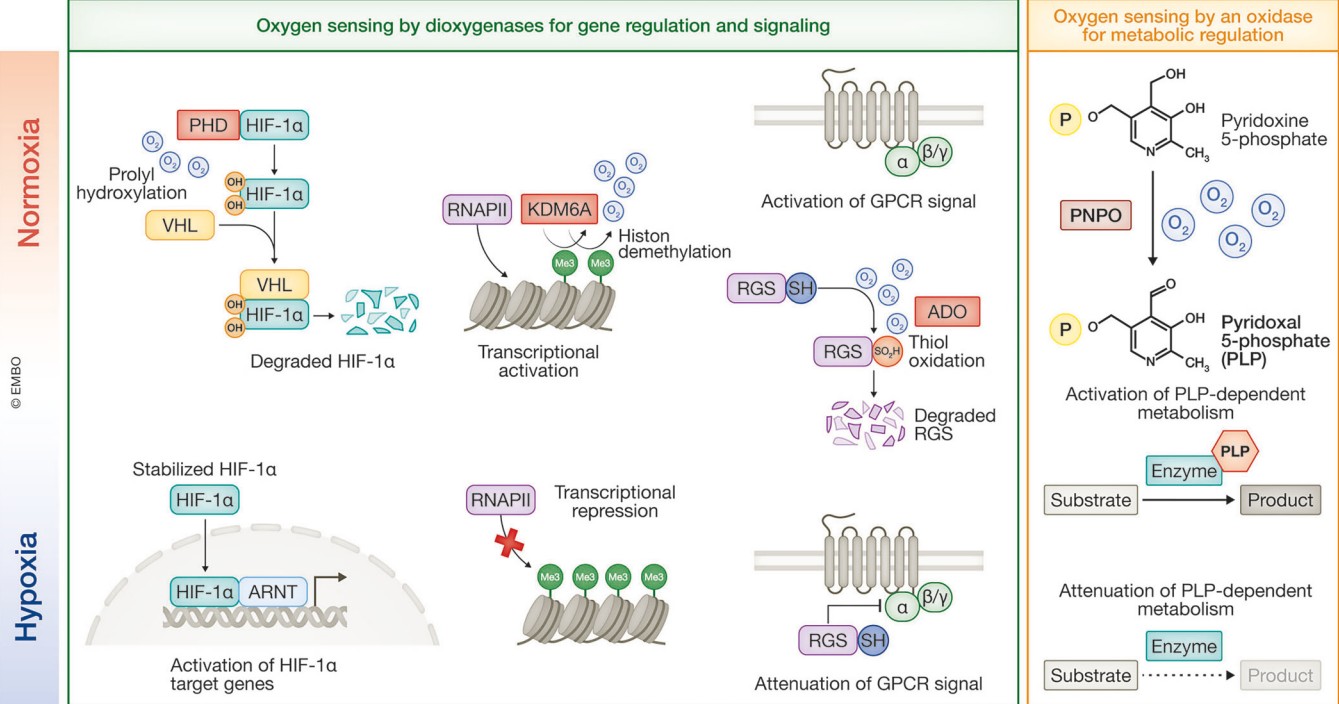

**Figure 8. O₂ sensing by dioxygenases and oxidases.**

PHD, KDM6A, and ADO are dioxygenases that regulate gene expression and signal transduction. In contrast, PNPO is an oxidase that modulates metabolism via PLP, leading to alterations in metabolite levels. P phosphate residue, Me3 trimethylated lysine residues of histones.

Interestingly, TET proteins, which require ferrous iron as an essential cofactor, have $K_m$ values for ferrous iron nearly 100 times higher than that of PHD2 (Laukka et al, 2016). This suggests that TET protein activities are highly susceptible to impairment under ferrous iron-limited conditions, such as during prolonged hypoxia. Consistently, in macrophages during inflammatory responses, prolonged hypoxia lessens ferrous iron concentrations (as a consequence of lysosomal inhibition) and abrogates TET2 protein accumulation, resulting in the delayed resolution of inflammatory response (Fig. 7). Of note, iron supplementation restores TET2 accumulation under conditions of lysosomal inhibition (Sekine et al, 2024). Based on these findings, we hypothesize that KDM5A, which also requires ferrous iron as a cofactor, may be vulnerable to ferrous iron depletion during hypoxia, if its $K_m$ value for ferrous iron is as high as those of TET proteins.

2-aminoethanethiol dioxygenase ADO), a member of the thiol dioxygenase family, was initially identified as a catalyst for taurine biosynthesis, converting cysteamine (2-aminoethanethiol) into hypotaurine (Dominy et al, 2007). Recent studies have revealed that ADO can accommodate N-terminal cysteines of polypeptides as substrates, oxidizing them to sulfinic acid (Masson et al, 2019). With a $K_m$ value for O₂ exceeding 500 μM, significantly higher than that of PHD proteins, ADO functions as an O₂ sensor. This oxidation marks proteins for degradation via the N-degron pathway in an oxygen-dependent manner. Notably, ADO has been shown to target regulators of G-protein signaling (RGS) proteins, which attenuate G-protein signaling by enhancing Gα-coupled GTP hydrolysis, and IL-32, a proinflammatory cytokine. These findings suggest that O₂ levels regulate G-protein signaling and that IL-32-mediated inflammation is exacerbated under hypoxic conditions.

With the exception of PNPO, all of the currently identified O₂ sensors are dioxygenases (Fig. 8). These dioxygenases regulate gene expression (e.g., PHD and KDM proteins) and signal transduction (e.g., ADO), indirectly causing metabolism to adapt to fluctuating O₂ levels. In contrast, PNPO, an oxidase, directly regulates metabolism by modulating PLP-dependent enzyme activities, subsequently altering gene expression. Given the large number of oxidases, it is likely that many remain unidentified with $K_m$ values for O₂ sufficiently high to enable them to act as O₂ sensors, directly regulating metabolism in response to O₂ availability.

The bacterial oxygen sensors that have been identified to date are not O₂-utilizing enzymes such as oxygenases and oxidases. One well-characterized example is the transcription factor fumarate-nitrate reduction (FNR) in *Escherichia coli*, which senses O₂ through the instability of its iron–sulfur cluster [4Fe-4S] (Fig. 9). In the presence of O₂, the [4Fe-4S] cluster bound to FNR is converted to [2Fe-2S], leading to the loss of DNA binding ability of FNR. Further exposure to O₂ gradually degrades the [2Fe-2S] cluster, resulting in a cluster-free FNR that can later incorporate a new [4Fe-4S] cluster (Crack et al, 2004; Green et al, 2014). Another example is the O₂-binding ability of ferrous-heme, which is utilized by the DosR/S/T (a three-component dormancy survival regulator) system in *Mycobacterium tuberculosis* for O₂ sensing (Fig. 9). DosS and DosT are heme-containing kinases that interact with O₂. In the absence of O₂, DosS and DosT exist in their active deoxy-heme forms, allowing them to autophosphorylate. The autophosphorylated DosS and DosT then transfer phosphate to DosR, activating the transcription of dormancy genes. Upon exposure to O₂,

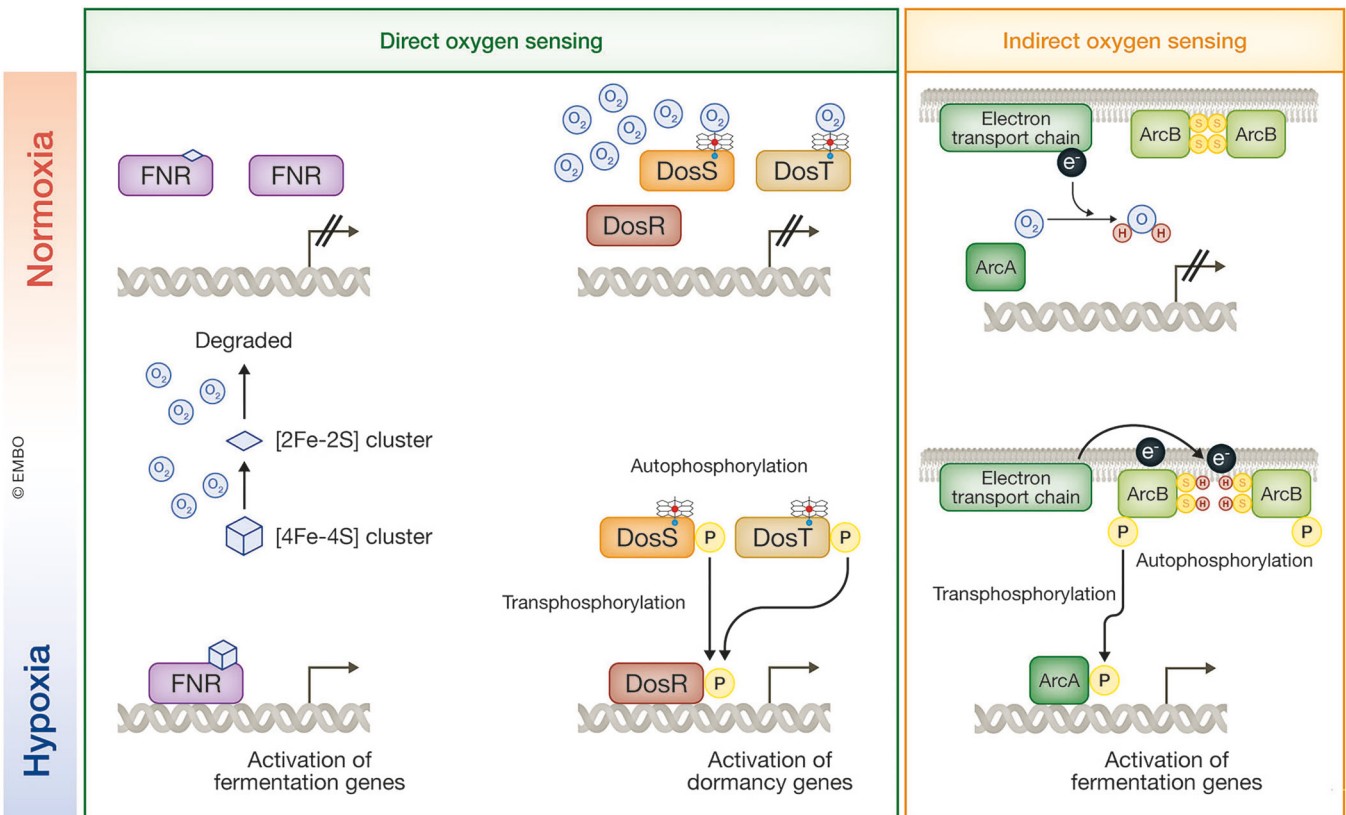

**Figure 9. O₂ sensing by bacterial system.**

FNR and DosS/T/R are direct oxygen-sensing systems, and ArcBA is an indirect oxygen-sensing system. Iron–sulfur cluster for FNR and heme for DosS/T are used for oxygen sensing, whereas ArcB utilizes cysteine thiols for oxygen sensing. P phosphate residue.

heme-bound $O_2$ induces conformational changes in DosS and DosT, inhibiting their kinase activities (Sousa et al, 2007; Green et al, 2014). While these systems directly sense $O_2$, the Arc (Anoxic Redox Control/ Aerobic Respiration Control) two-component regulatory system (ArcBA system) in *E. coli* functions as an indirect $O_2$ sensor (Fig. 9). The kinase ArcB contains two redox-active cysteine residues that are oxidized to form a disulfide bridge when terminal electron acceptors, including $O_2$, are available. In contrast, when these acceptors are absent, ArcB receives electrons, reducing its cysteine residues and activating its kinase function. This leads to autophosphorylation and subsequent transphosphorylation of ArcA. Phosphorylated ArcA then acts as a transcription factor to promote fermentation (Green et al, 2014; Brown et al, 2022). Notably, the ArcBA system appears to function as an oxygen-sensing mechanism that operates through competition between oxygen and sulfur for the role of electron acceptor.

# When oxygen-rich environment requires sulfur

## Sulfur-dependent antioxidant system

Under an oxidative atmosphere, managing oxygen toxicity is crucial for all organisms on Earth. Supersulfides play a significant

role as antioxidants due to their chemical properties; perthiols and polythiols have lower pKa values than simple thiols, enabling them to react readily with electrophiles and neutralize ROS (Ida et al, 2014; Benchoam et al, 2020; Barayeu et al, 2023). While activities with respect to ROS and radical species may vary among different supersulfide species, hydrogen peroxide, peroxynitrite and lipid radicals have been shown to be eliminated by GSSH, which is one of the most abundant supersulfides in cells. When comparing the reaction rate constants with GSSH, peroxynitrite reacts more than $10^4$-fold faster than $H_2O_2$ (Benchoam et al, 2020), suggesting that nucleophilic reactions by supersulfides exhibit selectivity for specific electrophiles. While kinetics are not fully quantified, lipid radicals are effectively quenched by GSSH as well as by CysSSH and inorganic hydropersulfides (Barayeu et al, 2023). These studies illustrate the fact that supersulfides can serve as essential biomolecules with antioxidant properties, and offer a primary defensive barrier in diverse species. In addition, organisms have also evolved inducible systems, mediated by transcriptional regulation, as secondary protective mechanisms against oxidative stress—mechanisms that also rely on sulfur.

In vertebrates, the KEAP1–NRF2 pathway is central to cytoprotection against oxidative stress (Yamamoto et al, 2018; Murakami et al, 2023) (Fig. 10). NRF2, encoded by the *NFE2L2* gene, is a potent transcriptional activator that regulates a broad array of cytoprotective genes, many of which encode enzymes

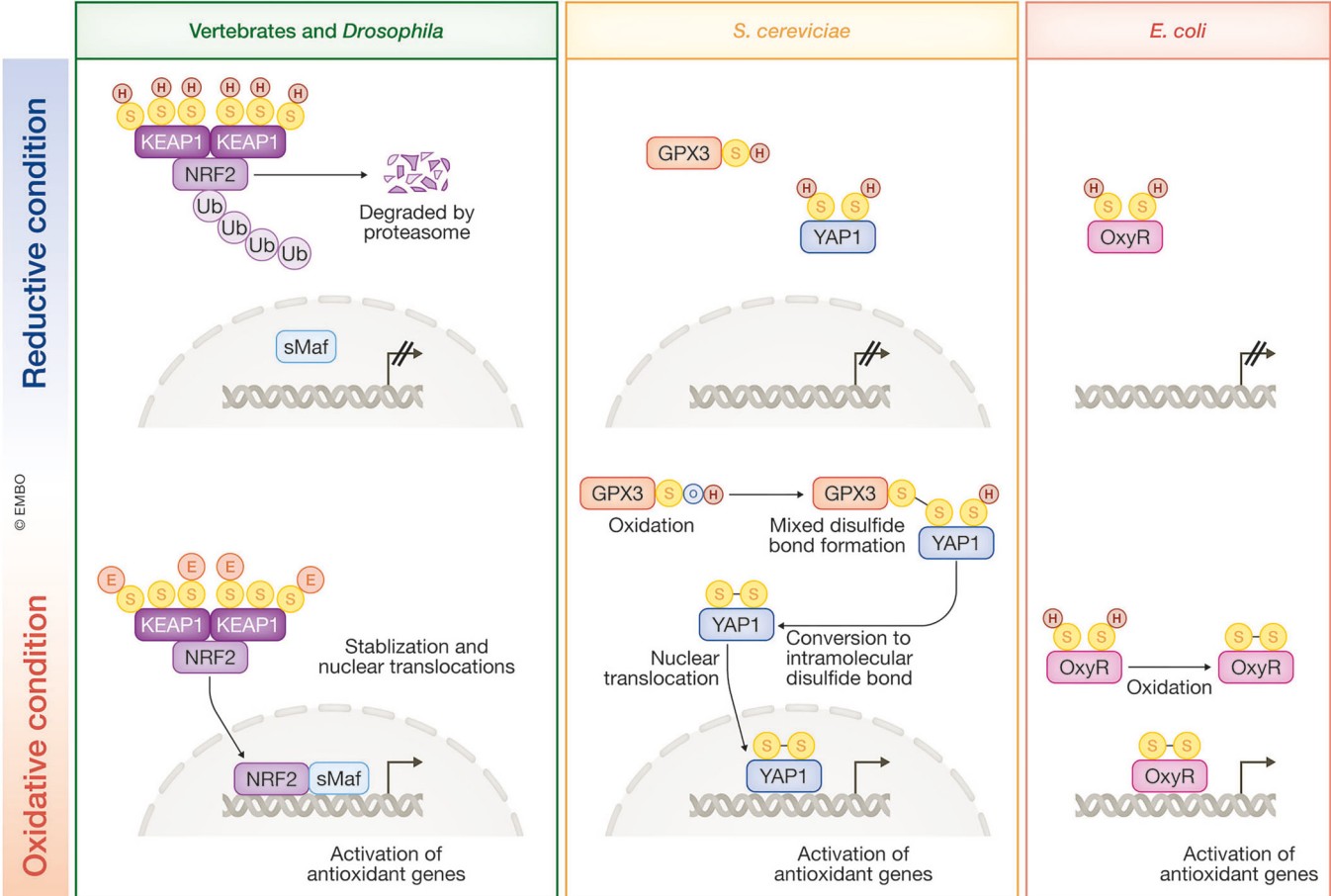

**Figure 10.  Antioxidant systems of various organisms.**

All these systems utilize cysteine thiols of sensor proteins for electrophile/ROS sensing. E electrophile, Ub ubiquitin.

involved in thiol-based redox reactions, including glutathione and thioredoxin metabolism. Notably, NRF2 regulates genes encoding enzymes responsible for supersulfide metabolism in a context-dependent manner. This includes CBS, CGL/CSE, and SQOR in mouse fibroblasts (Hourihan et al, 2013), thereby directly influencing supersulfide production in certain conditions. Moreover, by upregulating *Slc7a11*, which encodes a cystine transporter xCT, NRF2 enhances the cellular availability of cysteine, a key substrate for de novo supersulfide synthesis, and increases cellular levels of supersulfides (Alam et al, 2023). In this way, NRF2 plays a major role in governing sulfur metabolism and sulfur-dependent redox reactions.

KEAP1, a key regulator of NRF2, functions as a redox sensor through the reactivity of its thiol groups. KEAP1, as the substrate recognition subunit of a Cullin3-based ubiquitin E3 ligase complex, mediates the ubiquitination and proteasomal degradation of NRF2 under non-stress conditions, and keeps NRF2 activity low. KEAP1 contains multiple highly reactive cysteine residues that allow it to sense and respond to ROS and various electrophiles. Upon oxidative and electrophilic stress, these cysteine residues are modified, leading to KEAP1 inactivation, suppression of its E3 ligase activity, and subsequent induction of NRF2-mediated transcription. In addition to thiol oxidation leading to disulfide

bond formation by reactive oxygen species (ROS) (Suzuki et al, 2019), KEAP1 cysteine residues undergo S-alkylation by various endogenous and/or exogenous electrophiles (Wu and Papagiannakopoulos, 2020), such as succination by fumarate (Ooi et al, 2011; Adam et al, 2011) and S-guanylation by 8-nitro-cGMP (Sawa et al, 2007). Therefore, the KEAP1–NRF2 pathway exemplifies a sulfur-dependent cytoprotective mechanism against oxygen toxicity.

In *Drosophila*, the KEAP1–NRF2 pathway is conserved, with CNC serving as the NRF2 homolog and KEAP1 playing a similar role in oxidative stress defense. However, in *C. elegans*, the NRF2 homolog Skn-1 is present, but no KEAP1 homolog exists. Instead, Skn-1 activity is regulated by phosphorylation through the MAPK pathway (Inoue et al, 2005; Okuyama et al, 2010), with redox signals upstream of MAPK activation. For instance, oxidation or cysteine sulfenylation of IRE-1, an endoplasmic reticulum (ER) transmembrane protein critical for ER homeostasis, activates the p38 MAPK/SKN-1 antioxidant response. This pathway enhances stress resistance and promotes lifespan extension (Hourihan et al, 2016).

In unicellular organisms, the transcription factors YAP1 in *Saccharomyces cerevisiae* and OxyR in bacteria play pivotal roles in the oxidative stress response. Their cysteine residues are directly modified under oxidative stress, triggering the transcriptional

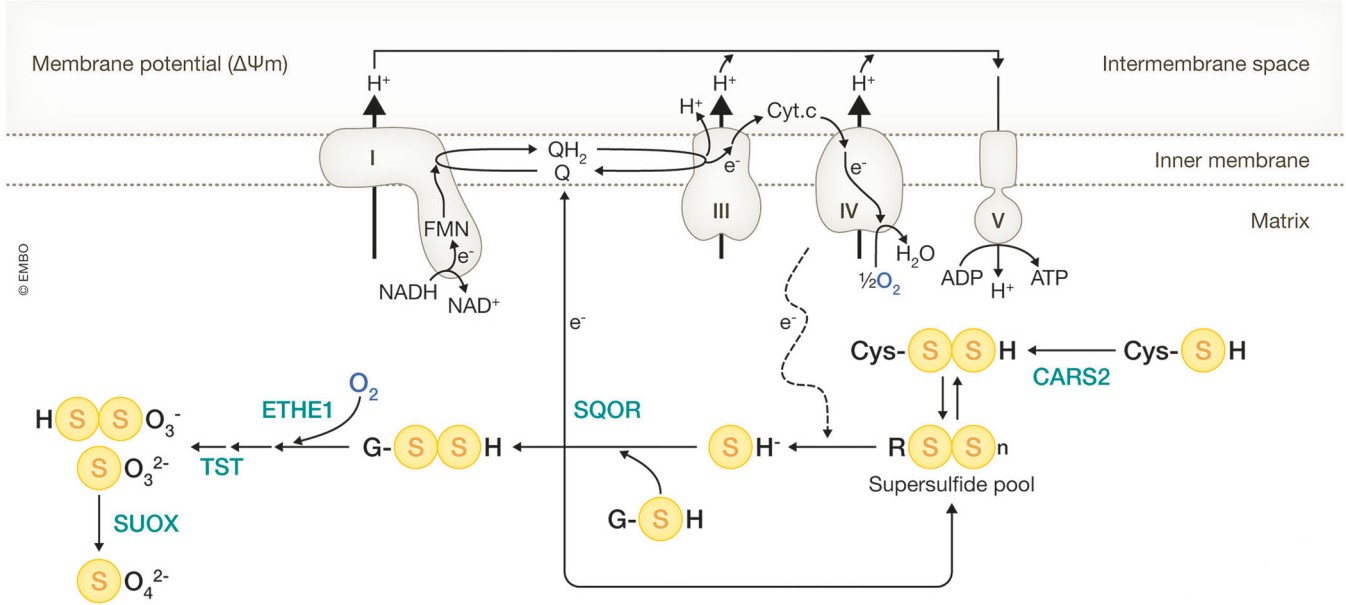

**Figure 11. Sulfur metabolism coupled with the mitochondrial ETC.**

Mitochondrial supersulfides participate in electron transfer, with some oxidized to sulfur oxides for excretion. Biorender software was used to create Figs. 1, 3, 4, 7, 8, 9, 10, and 11 under an academic license.

activation of target genes and enhancing resistance to oxidative damage (Ikner and Shiozaki, 2005; Hillion and Antelmann, 2015). YAP1 in *S. cerevisiae* translocates from the cytoplasm to the nucleus in response to oxidative stress, where it activates the transcription of antioxidant genes, including thioredoxin, thioredoxin reductase, glutathione reductase and γ-glutamylcysteine synthase (Fig. 10). YAP1 contains a nuclear localization signal (NLS) in its N-terminal region and a nuclear export signal (NES) in its C-terminal region. Under unstressed conditions, NES activity predominates, keeping YAP1 in the cytoplasm. Upon exposure to oxidative stress, a cysteine residue near the NES forms a mixed disulfide bond with GPX3 and subsequently an intramolecular disulfide bridge, likely masking the NES and facilitating YAP1 translocation into the nucleus (Ikner and Shiozaki, 2005). OxyR in *E. coli* forms a tetramer, with each subunit containing two redox-active cysteine residues (Fig. 10). In unstressed conditions, the reduced OxyR tetramer binds to DNA and functions as a repressor. Upon oxidative stress, an intramolecular disulfide bridge forms within each subunit, leading to oxidation of OxyR. The oxidized form of OxyR recognizes different DNA sequences compared to its reduced counterpart and facilitates the cooperative binding of RNA polymerase, thereby activating the transcription of a battery of antioxidant genes, including peroxiredoxin, catalase, thioredoxin, glutaredoxin, and glutathione reductase (Hillion and Antelmann, 2015). This direct, redox-sensitive mechanism enables unicellular organisms to rapidly adapt to fluctuating oxidative environments. Its simplicity and efficiency underscore the evolutionary benefit of directly coupling redox changes to gene expression. These mechanisms also shed light on how early life forms may have managed oxygen toxicity by using mechanisms that paved the way for more sophisticated oxidative stress defense systems in multicellular organisms, such as the KEAP1–NRF2 pathway, to emerge.

## Sulfur metabolism coupled with mitochondrial respiration

While $O_2$ serves as the terminal electron acceptor in the mitochondrial ETC under normoxic conditions, supersulfides have also been proposed as alternative electron acceptors in mitochondria, leading to the production of sulfide and inorganic supersulfides (Akaike et al, 2017; Alam et al, 2023; Matsunaga et al, 2023). CARS2, the mitochondrial isoform of cysteinyl tRNA synthetase, has been identified to possess cysteine persulfide synthase (CPERS) activity, catalyzing the conversion of two cysteine molecules into one cysteine persulfide and one serine or alanine. Mutations that reduce the CPERS activity of CARS2 result in decreased intracellular supersulfide levels, highlighting the significant role of CARS2 in supersulfide production (Akaike et al, 2017). Notably, CPERS activity-deficient CARS2 cannot sustain normal ETC activity, leading to mitochondrial membrane depolarization and reduced oxygen consumption. These findings underscore the critical importance of supersulfides in maintaining mitochondrial oxygen respiration (Akaike et al, 2017).

Mitochondria also harbor a sulfur oxidation pathway involving SQOR, ETHE1 (ethylmalonic encephalopathy protein 1, or persulfide dioxygenase), and SUOX (sulfite oxidase) (Hanna et al, 2023). Knockdown of CARS2, SQOR, or ETHE1 results in mitochondrial membrane depolarization, suggesting a functional coupling between supersulfide synthesis, sulfur oxidation, and the ETC that has enhanced mitochondrial energy metabolism (Alam et al, 2023). Mice that carry a *Cars2* mutation that specifically impairs CPERS activity exhibit reduced levels of supersulfides, such as CysSSH and hydropersulfide (HSSH), as well as sulfide (Matsunaga et al, 2023). Thus, supersulfides are likely to act as

electron acceptors, generating sulfide as a breakdown product. Here, sulfide is metabolized via the mitochondrial sulfur oxidation pathway, where it is converted to glutathione persulfide by SQOR and further processed into thiosulfate and sulfate through the coordinated actions of ETHE1, SUOX, and thiosulfate transferase (TST) (Fig. 11).

Iron–sulfur clusters are crucial components of electron carriers in the ETC, and their synthesis occurs within mitochondria. The process begins with NFS1, a cysteine desulfurase, which undergoes persulfidation through the desulfurization of cysteine, producing alanine as a byproduct (Parent et al, 2015; Gervason et al, 2019). This reaction marks the first step in iron–sulfur cluster synthesis. Supersulfides are formed as intermediates during the sulfur transfer from cysteine to the final assembly of iron–sulfur clusters. This also underscores the essential role of supersulfides in mitochondrial oxygen respiration. While the precise contribution of supersulfides to mitochondrial respiration remains an area for further investigation, it is evident that supersulfides play a fundamental role in supporting oxygen respiration in mitochondria. Their involvement highlights the intricate connection between sulfur metabolism and mitochondrial energy production.

# When using sulfur requires oxygen

## Generation of supersulfides under oxidative stress

Supersulfides are generated through multiple mechanisms. The de novo production of supersulfides is mediated by enzymes such as CARS, CBS, and CTH. CARS converts cysteine into cysteine persulfide (Akaike et al, 2017), while CBS and CTH utilize cystine to produce cysteine persulfide (Ida et al, 2014). Sulfane sulfur from cysteine persulfide is then transferred to other thiol- and per-/polythiol-containing molecules, resulting in the formation of various supersulfides. In addition, $H_2S$ is converted into persulfides by SQOR in mitochondria (Jackson et al, 2012). The absence of SQOR leads to sulfide accumulation, an increased NADH/NAD$^+$ ratio, and reduced oxygen consumption, indicating suppressed ETC function (Marutani et al, 2021). These effects resemble symptoms observed in patients who suffer from some mitochondrial diseases (Kanemaru et al, 2024). Beyond their enzymatic synthesis, supersulfides may also be generated non-enzymatically under oxidative stress (Paul and Snyder, 2015; Álvarez et al, 2017; Fukuto et al, 2018). During oxidative stress, thiols (–SH) can be oxidized to sulfenic acids (-SOH), which can react with $H_2S$ to form persulfide molecules (Wedmann et al, 2016). Since supersulfides, most typically hydropersulfides and hydropolysulfides, possess antioxidant properties, their non-enzymatic production under oxidative stress may function as an autoregulatory chemical circuit that contributes to redox balance regulation.

## PNPO–PLP system for connecting oxygen and sulfur

A common characteristic of all supersulfide-synthesizing enzymes is their dependency on PLP (see Fig. 5). PLP serves as a cofactor for various enzymes, primarily involved in amino acid metabolism, including transamination, decarboxylation, deamination, and racemization (Percudani and Peracchi, 2003). For example, δ-aminolevulinic acid synthase (ALAS), a rate-limiting enzyme in

heme synthesis, generates δ-aminolevulinic acid from succinyl-CoA and glycine in a PLP-dependent manner. PLP plays a critical role in one-carbon unit metabolism, catalyzed by SHMT2 (serine hydroxymethyltransferase 2), which converts serine into glycine and a one-carbon unit, and glycine dehydrogenase (glycine cleavage system P Protein; also known as GCSP), a subunit of the glycine cleavage system responsible for converting glycine into carbon dioxide and a one-carbon unit. An atypical example is glycogen phosphorylase, which relies on PLP to catalyze glycogen breakdown. Within redox regulation, PLP is indispensable for selenoprotein synthesis. It supports SEPSECS (Sep(O-Phosphoserine) tRNA:Sec (Selenocysteine) tRNA Synthase) in producing selenocysteine-tRNA and SCLY (Selenocysteine lyase) in decomposing selenocysteine. In this latter process, thioselenide (-S-SeH) is generated on the active-site cysteine of SCLY to recycle selenide (Omi et al, 2010). In addition, thioselenide is produced on PRDX6 via an inorganic selenide pathway that functions independently of PLP, further contributing to selenoprotein synthesis (Chen et al, 2024; Ito et al, 2024; Fujita et al, 2024).

Among various PLP-dependent enzymes, supersulfide-synthesizing enzymes are particularly sensitive to limitations in PLP availability (Sekine et al, 2024). During chronic hypoxia, sustained inhibition of PNPO activity leads to a gradual decline in intracellular PLP levels. This reduction compromises supersulfide production, subsequently inhibiting lysosomal acidification (see Fig. 7). Notably, the diminished lysosomal acidification observed under hypoxia can be reversed through supersulfide supplementation, indicating that the decrease in supersulfides suppresses lysosomal activity under low $O_2$ conditions. Oxygen plays a critical role in maintaining adequate supersulfide production, highlighting a connection between the PNPO–PLP pathway and sulfur metabolism regulation in response to $O_2$ levels.

From a phylogenetic perspective, PNPO homologs with pyridoxine 5′-phosphate oxidase activity, classified as PTHR10851 in the PANTHER database (https://pantherdb.org/), are widely distributed across many organisms but are absent in obligate anaerobic microorganisms (Mittenhuber, 2001; Denise et al, 2023). These anaerobes rely on a de novo PLP biosynthesis pathway that is independent of oxygen. The oxygen-dependent increase in supersulfide production via the PNPO–PLP pathway may have offered an evolutionary advantage by enhancing antioxidant capacity and mitigating oxygen toxicity. This coupling of oxygen utilization and supersulfide production underscores the critical role of the PNPO–PLP pathway in adapting to oxidative environments.

# Sulfur and oxygen for human health

## Oxidative stress versus reductive stress

Oxidative stress is a key factor in the development and progression of various pathological conditions, including atherosclerosis, chronic obstructive pulmonary disease (COPD), Alzheimer's disease, and cancer (Forman and Zhang, 2021). While ROS serve as essential signaling molecules under normal physiological conditions, oxidative stress arises when ROS are produced in excessive amounts and/or inappropriately generated at the wrong time or location. Excessive ROS causes the undesired oxidation of

macromolecules such as membrane lipids, structural proteins, enzymes, and nucleic acids, leading to impaired cellular function and cell death. Furthermore, aberrant ROS generation disrupts proper signal transduction, further contributing to cellular dysfunction. Excessive lipid oxidation, characterized by the accumulation of lipid radicals and peroxides, leads to a form of cell death known as ferroptosis (Dixon et al, 2012; Zheng and Conrad, 2025). Ferroptosis is a distinct form of cell death that occurs without executioner proteins, unlike other programmed cell death pathways such as apoptosis and pyroptosis, which depend on caspase-3 and gasdermin, respectively. Its uniqueness lies in its intricate connections to various metabolic processes that regulate cellular redox balance. Moreover, ferroptosis is implicated in numerous human pathologies associated with ROS (Berndt et al, 2024).

The KEAP1–NRF2 pathway plays a critical role in defending against oxidative stress. Insufficient NRF2 activity exacerbates ROS-related pathological conditions, whereas NRF2 activation mitigates them (Yamamoto et al, 2018; Murakami et al, 2023). For instance, NRF2 activation has been shown to protect against ischemia–reperfusion injury in organs such as the kidney, liver, heart, and brain (Nezu et al, 2017; Masuda et al, 2014; Katsumata et al, 2014; Alfieri et al, 2013). Similarly, noise-induced hearing loss, which shares mechanisms with ischemia–reperfusion injury in the cochlea, is closely linked to oxidative stress. During noise exposure, blood circulation in the cochlea is impaired due to vasoconstriction, and blood flow gradually recovers after the exposure, triggering ischemia–reperfusion injury. Studies in mouse models have demonstrated that NRF2 deficiency exacerbates noise-induced hearing loss, whereas NRF2 activation provides significant protection (Honkura et al, 2016). NRF2, with its antioxidant activity, also regulates iron metabolism, reducing lipid peroxidation and protecting against ferroptosis (Dodson et al, 2019; Anandhan et al, 2023).

NRF2 activation induces numerous $NAD^+$-consuming aldehyde dehydrogenases, leading to an increased cellular $NADH/NAD^+$ ratio and a redox imbalance known as NADH reductive stress. Persistent NRF2 activation, such as that caused by Cullin3 mutations, results in lipodystrophy in the liver and systemic metabolic alterations due to reductive stress (Gu et al, 2024). In humans, mutations in αB-crystallin are associated with cardiomyopathy caused by aberrant protein aggregate formation (Rajasekaran et al, 2007). These mutant αB-crystallin proteins sequester KEAP1 into aggregates, driving persistent NRF2 activation and exacerbating reductive stress-induced protein aggregation (Rajasekaran et al, 2011). Reductive stress has been shown to promote protein aggregation by impairing endoplasmic reticulum (ER) protein folding pathways, leading to cellular dysfunction and cell death, including impaired neurogenesis and myogenesis (Narasimhan et al, 2020; Rajasekaran et al, 2020). An interesting response mechanism to reductive stress has been reported. FNIP1 (Folliculin-Interacting Protein 1) functions as a sensor for reductive stress and plays a crucial role in regulating myogenesis. FNIP1 normally contains oxidized cysteine residues that stabilize it to limit mitochondrial activity. In reductive conditions, FNIP1 cysteine residues become reduced, triggering its ubiquitination by a CUL2-based E3 ligase for proteasomal degradation. This degradation activates mitochondria to produce ROS, helping to restore redox balance (Manford et al, 2020; Manford et al, 2021).

## Supersulfides for inflammation regulation

Inflammation is a key driver of numerous pathological conditions and contributes to the age-related functional decline of tissues and organs. The activation of NRF2, either through genetic inhibition of KEAP1 or pharmacological inhibition using electrophilic compounds, has demonstrated potent anti-inflammatory effects (Itoh et al, 2004; Kobayashi et al, 2016; Gopal et al, 2017; Suzuki et al, 2017; Mills et al, 2018). Electrophilic compounds such as dimethyl fumarate and itaconate achieve this by covalently modifying KEAP1 thiol groups, thereby inhibiting NRF2 ubiquitination and promoting NRF2 stabilization and pathway activation. The NRF2 activation results in upregulation of cytoprotective genes and downregulation of inflammatory genes. Interestingly, electrophiles also exhibit NRF2-independent anti-inflammatory properties. For example, dimethyl fumarate and itaconate modulate inflammatory pathways by targeting other molecules beyond KEAP1 (Schulze-Topphoff et al, 2016; Bambouskova et al, 2018). These dual mechanisms suggest a broader role for electrophiles in controlling inflammation through both NRF2-dependent and -independent pathways.

Supersulfides exhibit potent anti-inflammatory properties (Zhang et al, 2019; Takeda et al, 2023; Matsunaga et al, 2023). The synthetic supersulfide N-acetylcysteine tetrasulfide (NACS2) significantly suppresses LPS-induced proinflammatory responses in macrophages by inhibiting Toll-like Receptor 4 (TLR4) signaling, potentially acting as a supersulfide donor (Zhang et al, 2019). Endogenously produced supersulfides also contribute to the regulation of inflammation in macrophages. Upon LPS stimulation, the cystine transporter xCT, encoded by *Slc7a11*, is markedly upregulated, leading to increased supersulfide production. Notably, xCT-deficient macrophages exhibit heightened inflammatory responses, suggesting that endogenously produced supersulfides form a negative feedback loop to modulate macrophage inflammation (Takeda et al, 2023). Further supporting this notion, loss of function in the supersulfide-synthesizing enzyme CARS2 correlates with increased susceptibility to inflammatory insults and infections in pulmonary tissues, as observed in *Cars2* heterozygous mutant mice (Matsunaga et al, 2023). While the functional targets of supersulfides in inflammation regulation remain to be fully identified, proteins with altered supersulfidation in macrophages following LPS stimulation have been identified (Salti et al, 2024). These proteins are potential effectors mediating the anti-inflammatory effects of supersulfides. Similarly, in the context of antiviral activity, inorganic supersulfides target viral cysteine proteases, which are implicated in the inhibition of SARS-CoV-2 and influenza virus replication (Matsunaga et al, 2023).

Pathologically prolonged $O_2$ limitation exacerbates inflammation and delays its resolution. The $O_2$-sensitive enzyme PNPO plays a critical role in the bioactivation of vitamin B6, converting it into PLP. Under chronic hypoxia, PNPO activity is impaired, leading to a reduction in PLP levels and consequently a decrease in supersulfide production, especially in cells that depend on de novo synthesis of supersulfides (Sekine et al, 2024). Macrophages differentiated under hypoxic conditions exhibit diminished supersulfide levels and display an exaggerated and prolonged inflammatory response to LPS stimulation, attributed to the loss of TET2 function, a key mediator of inflammation resolution (see Fig. 7). Remarkably, supplementation with exogenous supersulfides under chronic hypoxic conditions effectively reverses the proinflammatory phenotype of these

macrophages, further supporting the anti-inflammatory role of supersulfides. Despite these findings, a critical question remains: could the inhibition of de novo supersulfide synthesis have adaptive benefits during physiologically prolonged hypoxia in vivo? For instance, in certain physiological or pathological contexts, a controlled inflammatory response might confer advantages, such as promoting tissue remodeling or pathogen clearance. Investigating the interplay between hypoxia, supersulfide metabolism, and inflammation in diverse in vivo models will be essential to fully understand the potential therapeutic implications of targeting supersulfide pathways in chronic hypoxic conditions.

## Redox imbalance in cancer

Cancer and redox imbalance are deeply interconnected, playing critical roles in tumorigenesis and cancer progression (Wu et al, 2024). Cancer cells, driven by their high metabolic demands to sustain rapid proliferation, produce significantly elevated levels of ROS compared to normal cells. These elevated ROS levels can be both deleterious and advantageous; while excessive ROS can cause cellular damage, moderate levels of ROS promote cancer cell survival, proliferation, and metastasis. For instance, ROS activate multiple pro-tumorigenic signaling pathways, including RAS–MAP, PI3K–AKT, and IKK–NFκB pathways (Wu et al, 2024). Notably, sulfur-containing residues often function as molecular switches that regulate protein activity in response to ROS. Oxidation of cysteine residues in EGFR enhances its kinase activity, leading to hyperactivation of growth signaling (Paulsen et al, 2011). Similarly, oxidation of cysteine residues in the tumor suppressor PTEN inhibits its activity, thereby activating the PI3K–AKT pathway (Lee et al, 2002). In addition, methionine oxidation in PKM2 stabilizes its active tetrameric form, promoting cell migration and metastasis (He et al, 2022). Once ROS levels exceed the threshold of their beneficial effects, cancer cells activate sophisticated antioxidant defense systems to mitigate oxidative damage and maintain redox homeostasis. These defenses include the well-established KEAP1–NRF2 pathway and emerging mechanisms linked to supersulfide metabolism (Hayashi et al, 2024).

Persistent activation of the NRF2 pathway is a common feature in several cancers, particularly those of the lung, esophagus, bladder, and head and neck regions (Härkönen et al, 2023). This aberrant activation often results from somatic mutations in *KEAP1* or *NFE2L2* (encoding NRF2), which disrupt the KEAP1–NRF2 regulatory axis. According to a recent pan-cancer whole genome analysis, non-small cell lung cancer has the highest mutation frequency in both *KEAP1* and *NFE2L2* (ICGC/TCGA Pan-Cancer Analysis of Whole Genomes Consortium, 2020). In liver cancers, the abnormal accumulation of p62, a cargo protein involved in selective autophagy, inhibits the KEAP1–NRF2 interaction, leading to the stabilization and activation of NRF2 (Saito et al, 2016). Similarly, in a subset of renal cancers associated with fumarate hydratase (FH) mutations, accumulated fumarate modifies KEAP1 thiols, further enhancing NRF2 activity (Ooi et al, 2011; Adam et al, 2011). These alterations significantly increase the antioxidant capacity of cancer cells, allowing them to survive under high oxidative stress conditions. This enhanced resilience contributes to resistance against chemotherapy and radiotherapy, ultimately resulting in poor clinical outcomes (Inoue et al, 2012; Onodera et al, 2014; Jeong et al, 2017). Consequently, NRF2 activation

represents a double-edged sword: while it protects normal cells from oxidative damage, its dysregulation in cancer cells fosters tumor survival and resistance to therapies (Kitamura and Motohashi, 2018). NRF2 further promotes cancer stemness (Okazaki et al, 2020), metastasis (Lignitto et al, 2019; Wiel et al, 2019), and immunoevasion (Marzio et al, 2022).

Elevated levels of supersulfides have been reported in ovarian, breast, pancreatic, and esophageal cancers (Honda et al, 2021; Erdélyi et al, 2021; Czikora et al, 2022; Asamitsu et al, 2025). Supersulfides in general exhibit strong antioxidant properties by readily reacting with and neutralizing radicals, including lipid radicals, which are abundantly generated in cancer cells due to heightened metabolic activity. This radical-quenching capacity effectively protects cancer cells from oxidative damage (Wu et al, 2022; Kaneko et al, 2022; Barayeu et al, 2023). Among the benefits conferred by elevated supersulfide production, their anti-ferroptotic effects are particularly significant, as they prevent lipid peroxidation-induced cell death, thereby supporting cancer cell survival and progression.

Persistently activated NRF2 under the sustained activation of growth signals drives metabolic reprogramming that redirects glucose and glutamine metabolism. This reprogramming results in increased production of NADPH, serine, purine nucleotides, and glutathione, which collectively support the aggressive tumorigenesis of NRF2-activated cancer cells (Mitsuishi et al, 2012; DeNicola et al, 2015). While these metabolic alterations benefit cancer cells, they are accompanied by trade-offs that create metabolic vulnerabilities. Exploiting these vulnerabilities, such as redox imbalances and associated features, has been proposed as an effective approach for developing cancer therapies. In normal redox regulation, oxidative shifts in the redox balance activate NRF2 by suppressing KEAP1, triggering the NRF2-mediated antioxidant pathway. This restores redox balance to a neutral state, leading to a decline in NRF2 activity. However, when NRF2 is forcibly activated—such as through KEAP1 inactivation—in normal cells, these cells are often excluded by cell competition (Hirose et al, 2023), likely due to their inability to sustain the metabolic burden required for maintaining a proliferative state. One major contributor to this burden is a shortage of glutamate, a trade-off incurred by enhancing antioxidant capacity (Romero et al, 2017; Sayin et al, 2017). NRF2 activation robustly upregulates several target genes, including SLC7A11 (which encodes the cystine transporter xCT) and GCLC and GCLM (which encode the two subunits of γ-glutamylcysteine synthetase, γGCL—the rate-limiting enzyme for glutathione synthesis). This upregulation increases glutamate demand because xCT, functioning as a cystine-glutamate antiporter, exports glutamate, and γGCL consumes glutamate as a substrate in glutathione synthesis. To survive under these conditions, cells must compensate by enhancing glutamine uptake and converting glutamine to glutamate. Cancer cells with persistent NRF2 activation adapt their metabolism by exploiting the glutamine transporter SLC1A5 for glutamine uptake and glutaminase for its conversion to glutamate (Romero et al, 2017; Sayin et al, 2017). Thus, while persistent NRF2 activation equips cancer cells with a robust defense against oxidative stress, it simultaneously necessitates metabolic adaptations to fulfill altered demands for specific metabolites and mitigate associated trade-offs (Romero et al, 2017; LeBoeuf et al, 2020; Ding et al, 2021). Reductive stress has also been identified as a vulnerability in NRF2-activated cancer cells

(Weiss-Sadan et al, 2023). Investigating the vulnerabilities of cancer cells that depend on supersulfides will offer intriguing and important opportunities for developing targeted anticancer therapies for such cancers.

# Future assignments and perspective

The words of Albert Szent-Györgyi, the 1937 Nobel Laureate, encapsulate the essence of life's processes: "Life is nothing but an electron looking for a place to rest" (Trefil et al, 2009). This statement highlights the fundamental role of redox reactions in all living organisms, underscoring the intricate interplay between sulfur and oxygen as a foundational basis for life. Advances in technology and analytical methods have revealed that supersulfides are biomolecules commonly occurring across species, reshaping our understanding of biological processes. By considering supersulfides, numerous molecular events that were previously unrecognized have come to light, expanding our knowledge of life's complexities. However, many questions remain unanswered. For instance, the precise molecular mechanisms through which sulfur supports mitochondrial function in aerobic organisms are still unclear. In addition, the dynamic changes in $O_2$ availability within living organisms and their impact on sulfur metabolism in physiological contexts remain critical areas of inquiry. These questions underscore the potential for groundbreaking discoveries at the intersection of sulfur and oxygen research. As our understanding deepens, new insights into the role of these elements in life processes are certain to emerge, offering exciting opportunities for scientific exploration and innovation.

# Peer review information

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

## Acknowledgements

We thank the members of the Department of Medical Biochemistry and the Department of Environmental Medicine and Molecular Toxicology at Tohoku University Graduate School of Medicine for continuous supports. This work was supported by JSPS (grant numbers 18H05277 (TA), 23H02672 (HS), 21H05263 (TA), 22K19397 (TA), 23K20040 (H.M and TA), 24H00063 (TA), 21H04799 (HM), 21H05258 (TA and HM), 21H05264 (HM) and 24H00605 (HM)), JST CREST Grant Number JPMJCR2024 (20348438 to TA), and AMED under grant number JP24gm6710026 (HS). The funders had no role in the study design, data collection and analysis, decision to publish or manuscript preparation.

## Author contributions

**Hiroki Sekine**: Conceptualization; Funding acquisition; Writing—original draft; Writing—review and editing. **Takaaki Akaike**: Conceptualization; Funding acquisition; Writing—review and editing. **Hozumi Motohashi**: Conceptualization; Supervision; Funding acquisition; Visualization; Writing—original draft; Writing—review and editing.

## Disclosure and competing interests statement

The authors declare no competing interests.

