## [Peer Review File · The EMBO Journal]

Oxygen Needs Sulfur, Sulfur Needs Oxygen: A Relationship of Interdependence

Hiroki Sekine, Takaaki Akaike, and Hozumi Motohashi

Corresponding author(s): Hozumi Motohashi (hozumi.motohashi.a7@tohoku.ac.jp), Hiroki Sekine (hiroki.sekine.c2@tohoku.ac.jp)

Review Timeline:

Submission Date:	5th Jan 25
Editorial Decision:	17th Feb 25
Revision Received:	10th Mar 25
Editorial Decision:	28th Mar 25
Revision Received:	29th Apr 25
Accepted:	30th Apr 25

Editor: William Teale

Transaction Report:

Dear Prof. Motohashi,

Thank you for submitting your review entitled "Oxygen Needs Sulfur, Sulfur Needs Oxygen: A Relationship of Interdependence" (EMBOJ-2024-119705) to The EMBO Journal. As you will see, both referees considered your work to be timely and interesting, making some specific constructive comments which I would like you to consider carefully.

Given the referees' positive recommendations, I would like to invite you to submit a revised version of the manuscript, addressing the comments of all three reviewers. Please also prepare a point-by point response that details how you have addressed these comments.

I will now send the figures to our artists so that they can adapt them to our house style. I will also go through your revised text myself and make any editorial suggestions which I think might improve the review. In addition, there are some remaining editorial points which need to be addressed. In this regard, would you please:

- include up to five keywords,
- rename the Conflict of Interests statement the 'Disclosure and Competing Interests Statement', and
- change the reference format so that (in the reference section) 'et al.' is used after 10 author names; DOIs should only be used for preprints and datasets that have not yet been published.

I look forward to receiving these changes. EMBO Press is an editorially independent publishing platform for the development of EMBO scientific publications.

Best wishes,

William Teale

William Teale, PhD
Editor
The EMBO Journal
w.teale@embojournal.org

We realize that it is difficult to revise to a specific deadline. In the interest of protecting the conceptual advance provided by the

work, we recommend a revision within 3 months (18th May 2025). Please discuss the revision progress ahead of this time with the editor if you require more time to complete the revisions. Use the link below to submit your revision:

Referee #1:

This excellent review, compiled by key dedicated experts in the field, provides an insightful exploration of the fundamental roles of oxygen and sulfur in biological systems, emphasizing their evolutionary significance and interplay in redox regulation. It highlights how organisms rely on sulfur species (including persulfides) and oxygen for energy metabolism, homeostasis, and adaptive responses to environmental stress. The manuscript is well-organized and offers a comprehensive review of recent progress in this emerging field. The reviewer has only a couple of minor suggestions for improvement.

- 1) Importantly, throughout the manuscript, several terms that were initially defined as abbreviations (e.g., ETC, PLP) and are later spelled out in full (or vice versa). Please ensure for consistency and first give full name and then use abbreviation.
- 2) The authors state that "supersulphides, such as hydrosulphides and hydropolysulphides, have strong antioxidant properties." Could the authors discuss the specific antioxidant features of persulfides in comparison to other endogenous (or even exogenous) antioxidants? In particular, is it known for which reactive oxygen species or radical species the persulphide has a high scavenging affinity?
- 3) Is there any known crosstalk between sulfur and oxygen metabolism at the level of gene regulation? For example, are persulfide synthesis enzymes (e.g., CBS, CSE, and CARS) regulated by NRF2-KEAP1 or HIF-PHD pathways? Additionally, the role of SLC7A11, which is upregulated by NRF2 and enhances persulfide production, would be worth discussing in this context.
- 4) Regarding oxygen-sensing pathways, do HIF-PHD and PNPO-PNPL have distinct roles depending on the duration of hypoxia (acute vs. chronic) or the tissue/cell type involved? Further description of this point would be valuable.
- 5) Can the authors specifically mention the identity of this 'metabolic load' caused by chronic NRF2 activation? Is it related to high cysteine burden or decreased glutamate levels due to SLC7A11 upregulation? Additional discussion or speculation with mentioning references would be an asset.
- 6) Page 8, first para: what does the number (2) refer to?
- 7) Page 16: Besides oxidative modification, can the authors also mention other PTMs on KEAP1?
- 8) Page 19, 2nd paragraph: "enzyme-bound thioselenide". Do the authors refer to PRDX6, which was recently identified by three groups independently to be the intracellular selenium carrier? Please include (PMID: 38867112; PMID: 39547222; PMID: 39547224).
- 9) Page 22: the Ref "S. Narasimhan" should be fixed.
- 10) Page 25, 1st para: Can the authors quickly specify how ROS would "promote cancer cell survival, proliferation and metastasis." This is a rather general statement and should be supported by some mechanistic details.
- 11) On the same page, lung cancer should be included as KEAP1 mutations are often linked to lung adenocarcinoma.
- 12) On page 26, when discussing the anti-ferroptotic properties of persulfides, a brief explanation of ferroptosis itself would be beneficial for clarity and helpful for the general reader.
- 13) On page 26, the manuscript states: "KEAP1 inactivation-in normal cells, these cells are often excluded by cell competition (Hirose et al., 2023), likely due to their inability to sustain the metabolic burden required for maintaining a proliferative state.
- 14) In one of the figures, DHODH and glycerol-3-phosphate dehydrogenase (GPD2) should localize to the inner mitochondrial membrane facing the intermembrane space.

Referee #2:

This review is very interesting and detailed regarding the relationship of sulfur and oxygen in cellular metabolism, signal transduction and regulation, especially the role of such supersulfide species in mitochondrial respiration, signaling, cancer and human health and their connection to oxidative stress.

Overall, I have only few minor comments regarding some statements for chemolithotrophic sulfur bacteria and photosynthetic bacteria. Also some systems used for oxygen and sulfur sensing in bacteria should be also included in this review.

Major comments:

Page 3: The entire overview has no references, which is not acceptable. Please include references for all statements throughout the manuscript, including overview section.
Please introduce all abbreviations first, such as oxygen (O₂), KEAP1-Nrf2, PNPO-PLP etc. and then use the abbreviations throughout the manuscript.

Page 3, line 4: change to "primitive anoxygenic phototrophic bacteria"

Page 3, line 8: change to "This led to evolution of aerobic organisms that utilized oxygen for respiration as the best terminal electron acceptor"

Page 3, line 8-9: This sentence is not correct, since many chemolithotrophic sulfur bacteria at such hydrothermal vents are aerobic symbionts and use oxygen as electron acceptor, other chemolithotrophs are facultative aerobic and can use nitrate or oxygen, depending on the habitats. Please correct and cite the literature carefully when making such statements.

Page 5: Here again the authors state that "elemental sulfur and sulfate acting as electron acceptors", which is wrong. Elemental sulfur S⁰ is formed upon oxidation of the electron donor sulfide (H₂S) as intermediate. Elemental Sulfur is stored in granula by many sulfur bacteria and used as electron donor when sulfide is limiting. Then sulfur is oxidized to sulfate. Thus, elemental sulfur is an electron donor and no electron acceptor. As stated above most chemolithotrophs use oxygen as electron acceptor even these sulfur bacteria, which are endosymbionts at the hydrothermal vents rely on oxygen for respiration. Please correct these statements.

Page 6: Also this statement is somehow misleading and not very professional written: "This implies that early phototrophic bacteria with primitive photosynthetic systems may have found it easier to extract electrons from hydrogen sulfide rather than water"

Basically, the anoxygenic phototrophs have just one photosystem (PSI or PSII) and evolved in an anaerobic atmosphere before the evolution of oxygen. Thus, they used other electron donors, such as H₂S for electron transport in the energy metabolism. Cyanobacteria evolved two PS systems (PSI and PSII for water splitting) to produce oxygen 2,7 billion years ago, which was the basis for evolution of aerobic life.

Page 6: SqrR: It would be required to indicate which genes that are regulated by the repressor and what is the functions of the SqrR regulon under sulfide-rich conditions, when the genes are expressed. Please write more details. Also the SqrR repressor and its functions in sulfur metabolism has been studied very detailed in other aerobic model bacteria, which should be mentioned as well.

Page 6: Please explain more detailed what is regulated by sulfate transporters Sul1 and Sul2 as sulfate sensors.

Page 12: While the oxygen-sensing regulatory factors and enzymes are described in detail for eukaryotic organisms, there are no examples of bacterial oxygen sensors mentioned. The authors might include few prominent examples of bacterial oxygen sensors as well, such as the ArcBA system and FNR regulator of E. coli, since they are also redox-regulated.

Page 16: The OxyR and Yap1 regulatory mechanisms and functions of the controlled genes should be more detailed described.

Page 23: Two references have only doi numbers in the Ref-list and must be updated with full citation.

Figure 2 should be more detailed explained in the legend as mentioned above in my comments for the text. The figure is not very professional for some terms and need to corrected:

Please change "photosynthetic bacteria" to "anoxygenic phototrophic bacteria"

Please change primitive to "PSI or PSII" and advanced to "PSI and PSII"

Response to reviewers' comments.

Referee #1:

This excellent review, compiled by key dedicated experts in the field, provides an insightful exploration of the fundamental roles of oxygen and sulfur in biological systems, emphasizing their evolutionary significance and interplay in redox regulation. It highlights how organisms rely on sulfur species (including persulfides) and oxygen for energy metabolism, homeostasis, and adaptive responses to environmental stress. The manuscript is well-organized and offers a comprehensive review of recent progress in this emerging field. The reviewer has only a couple of minor suggestions for improvement.

We are very happy to know that the reviewer highly appreciates our review article. We thank the reviewer for encouraging comments.

1) Importantly, throughout the manuscript, several terms that were initially defined as abbreviations (e.g., ETC, PLP) and are later spelled out in full (or vice versa). Please ensure for consistency and first give full name and then use abbreviation.

As suggested by the reviewer, we have defined abbreviations by spelling them out at their first occurrence and have used the abbreviations consistently thereafter.

2) The authors state that "supersulphides, such as hydrosulphides and hydropolysulphides, have strong antioxidant properties." Could the authors discuss the specific antioxidant features of persulfides in comparison to other endogenous (or even exogenous) antioxidants? In particular, is it known for which reactive oxygen species or radical species the persulphide has a high scavenging affinity?

We thank the reviewer for an important comment.

To clarify the distinction between supersulfides and simple thiol-containing molecules, we have provided an explanation using GSSH and GSH (page 9, lines 5 - 9). Additionally, we have discussed hydrogen peroxide, peroxyxynitrite, and lipid radicals, highlighting the high scavenging activity of supersulfides against these reactive species (page 18, lines 6 - 16).

3) Is there any known crosstalk between sulfur and oxygen metabolism at the level of gene regulation? For example, are persulfide synthesis enzymes (e.g., CBS, CSE, and CARS) regulated by NRF2-KEAP1 or HIF-PHD pathways? Additionally, the role of SLC7A11, which is upregulated by NRF2 and enhances persulfide production, would be worth discussing in this context.

As suggested by the reviewer, we have mentioned CBS, CGL/CSE and SQOR as context-dependent NRF2 target genes. Additionally, we have added a description of SLC7A11, which is regulated by NRF2 and contributes to supersulfide production (page 18, line 2 from the bottom - page 19, line 6).

4) Regarding oxygen-sensing pathways, do HIF-PHD and PNPO-PNPL have distinct roles depending on the duration of hypoxia (acute vs. chronic) or the tissue/cell type involved? Further description of this point would be valuable.

We thank the reviewer for raising this critical question. We have added a discussion on the

impact of acute and chronic hypoxia on the PHD-HIF pathway and the PNPO-PLP pathway, highlighting their functional differences based on the hypoxic duration (page 15, lines 12 - 16).

5) *Can the authors specifically mention the identity of this 'metabolic load' caused by chronic NRF2 activation? Is it related to high cysteine burden or decreased glutamate levels due to SLC7A11 upregulation? Additional discussion or speculation with mentioning references would be an asset.*

Following the reviewer's suggestion, we have addressed the metabolic burden caused by persistent activation of NRF2, highlighting glutamine shortage as a key issue (page 31, lines 10 - 34).

6) *Page 8, first para: what does the number (2) refer to?*

"Page 8, number (2)" (now page 10, line 6) means the second section ((2) PNPO-PLP system for connecting oxygen and sulfur) of chapter "5. Oxygen requirement for sulfur utilization".

7) *Page 16: Besides oxidative modification, can the authors also mention other PTMs on KEAP1?*

Following the reviewer's advice, we have mentioned post-translational modifications of KEAP1, specifically, S-alkylation, succination and S-guanylation (page 19, lines 14 - 18).

8) *Page 19, 2nd paragraph: "enzyme-bound thioselenide". Do the authors refer to PRDX6, which was recently identified by three groups independently to be the intracellular selenium carrier? Please include (PMID: 38867112; PMID: 39547222; PMID: 39547224).*

The phrase "enzyme-bound thioselenide" in the original manuscript referred to SCLY, not PRDX6. We apologize for the lack of clarity in our description. To clarify this, we have added a sentence explicitly stating that SCLY carries thioselenide.

Additionally, we appreciate the reviewer's suggestion regarding the relevant papers on PRDX6, which provide valuable insights. We have incorporated a reference to PRDX6 by citing these three papers (page 23, line 4 from the bottom - page 24, line 4).

9) *Page 22: the Ref "S. Narasimhan" should be fixed.*

The reference has been corrected. We thank the reviewer for pointing out.

10) *Page 25, 1st para: Can the authors quickly specify how ROS would "promote cancer cell survival, proliferation and metastasis." This is a rather general statement and should be supported by some mechanistic details.*

Following the reviewer's suggestion, we have provided three examples illustrating how ROS promote cancer cell survival, proliferation and metastasis (page 29, lines 7 - 18).

11) *On the same page, lung cancer should be included as KEAP1 mutations are often linked to lung adenocarcinoma.*

As suggested by the reviewer, we have mentioned non-small cell lung cancer (page 29, lines 1 - 4 from the bottom).

12) On page 26, when discussing the anti-ferroptotic properties of persulfides, a brief explanation of ferroptosis itself would be beneficial for clarity and helpful for the general reader.

As suggested by the reviewer, we have added a brief explanation of ferroptosis (page 25, line 11 - 16).

13) On page 26, the manuscript states: "KEAP1 inactivation-in normal cells, these cells are often excluded by cell competition (Hirose et al., 2023), likely due to their inability to sustain the metabolic burden required for maintaining a proliferative state.

Following the reviewer's advice, we have elaborated on the "metabolic burden" associated with glutamate regulation (see our response to Question No. 5) (page 31, lines 10 - 34).

14) In one of the figures, DHODH and glycerol-3-phosphate dehydrogenase (GPD2) should localize to the inner mitochondrial membrane facing the intermembrane space.

We thank the reviewer for an important comment. We have corrected Figure 5 (corresponding to Figure 6 in the revised manuscript).

Referee #2:

This review is very interesting and detailed regarding the relationship of sulfur and oxygen in cellular metabolism, signal transduction and regulation, especially the role of such supersulfide species in mitochondrial respiration, signaling, cancer and human health and their connection to oxidative stress.

Overall, I have only few minor comments regarding some statements for chemolithotrophic sulfur bacteria and photosynthetic bacteria. Also some systems used for oxygen and sulfur sensing in bacteria should be also included in this review.

We appreciate constructive comments from the reviewer.

Major comments:

Page 3: The entire overview has no references, which is not acceptable. Please include references for all statements throughout the manuscript, including overview section.

Following the reviewer's advice, we have cited appropriate papers in overview section (pages 3-4).

Please introduce all abbreviations first, such as oxygen (O₂), KEAP1-Nrf2, PNPO-PLP etc. and then use the abbreviations throughout the manuscript.

As suggested by the reviewer, we have defined abbreviations by spelling them out at their first occurrence and have used the abbreviations consistently thereafter.

Page 3, line 4: change to "primitive anoxygenic phototrophic bacteria"

We have changed the phrase as suggested (page 3, line 5).

Page 3, line 8: change to "This led to evolution of aerobic organisms that utilized oxygen for respiration as the best terminal electron acceptor"

We have changed the sentence as suggested (page 3, lines 9-10).

Page 3, line 8-9: This sentence is not correct, since many chemolithotrophic sulfur bacteria at such hydrothermal vents are aerobic symbionts and use oxygen as electron acceptor; other chemolithotrophs are facultative aerobic and can use nitrate or oxygen, depending on the habitats. Please correct and cite the literature carefully when making such statements.

This reviewer's comment likely refers to page 5, lines 8-9.

We appreciate the feedback and apologize for any ambiguity in our description. We intended to refer specifically to hydrothermal chimneys, which are anoxic, hot, and rich in reducing chemicals, when discussing hydrothermal vents. However, we recognize that our wording may not have fully conveyed this distinction.

To enhance clarity, we have revised the text to explicitly describe the four distinct hydrothermal vent habitats, namely, hydrothermal chimneys, the subsurface surrounding vents, vent-associated fauna, and hydrothermal plumes, as reviewed in Dick (2019). Additionally, we have expanded our explanation of hydrothermal environments, including both aerobic subsurface regions and anoxic hydrothermal chimneys, as well as the microorganisms adapted to these conditions (page 5, line 12 - page 6, line 10).

Page 5: Here again the authors state that "elemental sulfur and sulfate acting as electron acceptors", which is wrong. Elemental sulfur S^0 is formed upon oxidation of the electron donor sulfide (H_2S) as intermediate. Elemental Sulfur is stored in granula by many sulfur bacteria and used as electron donor when sulfide is limiting. Then sulfur is oxidized to sulfate. Thus, elemental sulfur is an electron donor and no electron acceptor. As stated above most chemolithotrophs use oxygen as electron acceptor even these sulfur bacteria, which are endosymbionts at the hydrothermal vents rely on oxygen for respiration. Please correct these statements.

We appreciate the reviewer's comment and again apologize for any lack of clarity in our description of hydrothermal vents. Among the four types of hydrothermal vent habitats, hydrothermal chimneys are particularly anoxic and hot. Under such extreme conditions, bacteria, especially thermophilic anaerobes, utilize elemental sulfur as an electron acceptor or reduce elemental sulfur for energy metabolism (Inagaki et al., 2004; Nakagawa et al., 2005; Campbell et al., 2006; Jelen et al., 2018; Dick, 2019). From a chemical perspective, elemental sulfur reacts with reduced glutathione in vitro to form glutathione persulfide, representing a reduction process of elemental sulfur. In anoxic environments, elemental sulfur serves as an electron acceptor in sulfur respiration. Conversely, in habitats where

oxygen is present, elemental sulfur can act as an electron donor, with oxygen serving as the electron acceptor.

To address this point, we have revised the text to explicitly clarify the dual roles of elemental sulfur, as both an electron donor and an electron acceptor, depending on the specific conditions within hydrothermal vent habitats (page 5, line 12 - page 6, line 10).

Page 6: Also this statement is somehow misleading and not very professional written: "This implies that early phototrophic bacteria with primitive photosynthetic systems may have found it easier to extract electrons from hydrogen sulfide rather than water" Basically, the anoxygenic phototrophs have just one photosystem (PSI or PSII) and evolved in an anaerobic atmosphere before the evolution of oxygen. Thus, they used other electron donors, such as H₂S for electron transport in the energy metabolism.

Cyanobacteria evolved two PS systems (PSI and PSII for water splitting) to produce oxygen 2,7 billion years ago, which was the basis for evolution of aerobic life.

We thank the reviewer for constructive comments.

Following the reviewer's comment, we have modified the description and mentioned PSI and PSII (page 6, line 14 - page 7, line 3).

Page 6: SqrR: It would be required to indicate which genes that are regulated by the repressor and what is the functions of the SqrR regulon under sulfide-rich conditions, when the genes are expressed. Please write more details. Also the SqrR repressor and its functions in sulfur metabolism has been studied very detailed in other aerobic model bacteria, which should be mentioned as well.

We thank the reviewer for constructive comments.

Following the reviewer's comment, we have provided a more detailed description of SqrR and expanded the discussion to include BigR and FisR (page 7, line 11 - page 8, line 7). Additionally, we have created a new figure (Figure 3) summarizing these sulfur sensor molecules.

Page 6: Please explain more detailed what is regulated by sulfate transporters Sul1 and Sul2 as sulfate sensors.

Following the reviewer's advice, we have expanded the explanation of Sul1 and Sul2, including their downstream signaling pathways (page 8, lines 11 - 16).

Page 12: While the oxygen-sensing regulatory factors and enzymes are described in detail for eukaryotic organisms, there are no examples of bacterial oxygen sensors mentioned. The authors might include few prominent examples of bacterial oxygen sensors as well, such as the ArcBA system and FNR regulator of E. coli, since they are also redox-regulated.

Following the reviewer's advice, we have mentioned FNR, DosR/S/T and ArcBA systems as bacterial oxygen sensing system (page 17, line 2 - line 1 from the bottom).

Page 16: The OxyR and Yap1 regulatory mechanisms and functions of the controlled genes should be more detailed described.

Following the reviewer's advice, we have explained OxyR and YAP1 regulatory mechanisms in more detail (page 20, line 7- line 4 from the bottom).

Page 23: Two references have only doi numbers in the Ref-list and must be updated with full citation.

We apologize for this oversight. As these doi numbers had already been incorporated into the reference list, we have deleted them.

*Figure 2 should be more detailed explained in the legend as mentioned above in my comments for the text. The figure is not very professional for some terms and need to corrected:
Please change "photosynthetic bacteria" to "anoxygenic phototrophic bacteria"
Please change primitive to "PSI or PSII" and advanced to "PSI and PSII"*

We thank the reviewer for an important advice. The Figure 2 and its legend has been revised according to the comment from the reviewer.

Dear Prof. Motohashi,

Thank you for submitting your manuscript for consideration by the EMBO Journal.

Given the referees' positive recommendations, I would like to invite you to submit a revised version of the manuscript, addressing the comments of all three reviewers. I should add that it is EMBO Journal policy to allow only a single round of revision, and acceptance of your manuscript will therefore depend on the completeness of your responses in this revised version.

We generally allow three months as standard revision time. As a matter of policy, competing manuscripts published during this period will not negatively impact on our assessment of the conceptual advance presented by your study. However, we request that you contact the editor as soon as possible upon publication of any related work, to discuss how to proceed.

Thank you for the opportunity to consider your work for publication. I look forward to your revision.

Yours sincerely,

William Teale

William Teale, PhD
Editor
The EMBO Journal
w.teale@embojournal.org

We realize that it is difficult to revise to a specific deadline. In the interest of protecting the conceptual advance provided by the work, we recommend a revision within 3 months (26th Jun 2025). Please discuss the revision progress ahead of this time with

the editor if you require more time to complete the revisions. Use the link below to submit your revision:

All editorial and formatting issues were resolved by the authors.

Dear Hozumi,

I am pleased to inform you that your manuscript has been accepted for publication in the EMBO Journal.

Congratulations on what I hope will be a really useful review!

Yours sincerely,

William

William Teale, PhD
Editor
The EMBO Journal
w.teale@embojournal.org
